# An Efficient High-Dimensional Gradient Estimator for Stochastic Differential Equations

**Shengbo Wang**
MS&E
Stanford University
Stanford, CA 94305
shengbo.wang@stanford.edu

**Jose Blanchet**
MS&E
Stanford University
Stanford, CA 94305
jose.blanchet@stanford.edu

**Peter Glynn**
MS&E
Stanford University
Stanford, CA 94305
glynn@stanford.edu

## Abstract

Overparameterized stochastic differential equation (SDE) models have achieved remarkable success in various complex environments, such as PDE-constrained optimization, stochastic control and reinforcement learning, financial engineering, and neural SDEs. These models often feature system evolution coefficients that are parameterized by a high-dimensional vector $\theta \in \mathbb{R}^n$, aiming to optimize expectations of the SDE, such as a value function, through stochastic gradient ascent. Consequently, designing efficient gradient estimators for which the computational complexity scales well with $n$ is of significant interest. This paper introduces a novel unbiased stochastic gradient estimator–the generator gradient estimator–for which the computation time remains stable in $n$. In addition to establishing the validity of our methodology for general SDEs with jumps, we also perform numerical experiments that test our estimator in linear-quadratic control problems parameterized by high-dimensional neural networks. The results show a significant improvement in efficiency compared to the widely used pathwise differentiation method: Our estimator achieves near-constant computation times, increasingly outperforms its counterpart as $n$ increases, and does so without compromising estimation variance. These empirical findings highlight the potential of our proposed methodology for optimizing SDEs in contemporary applications.

## 1 Introduction

We consider a family of jump diffusions $\left\{ X_\theta^x(t,s) \in \mathbb{R}^d : s \in [t,T] \right\}$ that are generated by stochastic differential equations (SDEs) and indexed by the initial condition $x \in \mathbb{R}^d$ at time $s$ and a parameter $\theta \in \Theta \subset \mathbb{R}^n$. In modern applications, the parameter $\theta$, encoding characteristics of an engineering model, often represents the weights of a deep neural network. This paper focuses particularly on scenarios where the dimension $n$ of $\theta$ is significantly greater than the dimension $d$ of the space. This setting naturally arises in the implementation of large AI architectures in modern applications.

38th Conference on Neural Information Processing Systems (NeurIPS 2024).

Concretely, for each $1 \leq i \leq d$, the $i$'th entry of $X_\theta^x(t, \cdot)$, denoted by $X_{\theta,i}^x(t, \cdot)$, satisfies the Itô SDE:

$$
\begin{aligned}
X_{\theta,i}^x(t, s) = x_i &+ \int_t^s \mu_{\theta,i}(r, X_\theta^x(t, r))dr \\
&+ \int_t^s \sum_{k=1}^{d'} \sigma_{\theta,i,k}(r, X_\theta^x(t, r-))dB_k(r) + \int_t^s dJ_{\theta,i}(r)
\end{aligned}
\tag{1.1}
$$

Here, $\{\mu_{\theta,i} : 1 \leq i \leq d\}$ and $\{\sigma_{\theta,i,k} : 1 \leq i, k \leq d\}$ are the drift and volatility, respectively, satisfying suitable regularity conditions (to be discussed). For simplicity in our introductory explanations, we will assume that the jump term $J_\theta$ is zero. However, incorporating this jump feature is valuable in many applied settings, and arises in various fields such as financial engineering [17], stochastic control [6], and neural SDE models [7]. Accordingly, we will fully integrate and discuss the jump components in our main results in Section 3.

The primary objective of this paper is to develop an efficient gradient estimator, with respect to $\theta$, for a large class of path-dependent expectations derived from an SDE. Concretely, we consider

$$
v_\theta(t, x) = E \left[ \int_t^T \rho_\theta(s, X_\theta^x(t, s))ds + g_\theta(X_\theta^x(t, T)) \right].
\tag{1.2}
$$

The value $v_\theta(t, x)$ represents the expected cumulative reward running $X_\theta^x$ from time $t$ to $T$. Here, $\rho_\theta$ and $g_\theta$ represents the reward rate and the terminal reward, respectively. This formulation encompasses a wide range of science and engineering problems including PDE-constrained optimization [22, 20], stochastic control and reinforcement learning [8], and neural SDE models [23].

The gradient $\nabla_\theta v_\theta(t, x) = (\partial_{\theta_1} v_\theta(t, x), \ldots, \partial_{\theta_n} v_\theta(t, x)) \in \mathbb{R}^n$ is of significant interest in the sensitivity analysis, learning, and optimization of these models. In particular, finding an efficient unbiased estimator for $\nabla_\theta v_\theta(t, x)$ with low variance is essential if one is to apply stochastic gradient descent to find near optimal policies or model parameters within the parametric class $\theta \in \Theta$.

Under reasonable smoothness and integrability conditions, it is natural to consider the pathwise differentiation estimator obtained by applying infinitesimal perturbation analysis (IPA) to the sample path of $X_\theta^x$ w.r.t. the $i$th coordinate of $\theta$. For instance, if $\rho_\theta(\cdot) = \rho(\cdot)$ independent of $\theta$ and $g = 0$, then we have a representation

$$
\partial_{\theta_i} v_\theta(0, x) = E \left[ \int_0^T \sum_{j=1}^d \partial_{x_j} \rho(t, X_\theta^x(t)) \partial_{\theta_i} X_{\theta,j}^x(t)dt \right].
\tag{1.3}
$$

where $\partial_{\theta_i} X_\theta^x(t)$ is the pathwise derivative of the process $X_\theta^x$ w.r.t. $\theta_i$. The processes $\{X_{\theta,j}^x, \partial_{\theta_i} X_{\theta,j}^x : i = 1, \ldots, n; j = 1, \ldots, d\}$ satisfy a system of $d + d \cdot n$ SDEs [11, Equation (3.31)], which must be jointly simulated. Therefore, to estimate the gradient, the pathwise differentiation method requires simulating this $d + d \cdot n$ dimensional SDE. Note that the dimension is linear in $n$, the dimension of the parameter space. Contemporary applications of SDEs in physics-informed and data-driven environments such as deep neural SDEs and deep RL where overparameterization excel, necessitate a model with exceptionally large $n$ that is often many orders of magnitude larger than $d$. Hence, simulating the SDE of dimension $d + d \cdot n$ becomes extremely resource-intensive. Motivated by these applications, we ask the following question:

*Can we device an efficient, unbiased, and finite variance estimator*
*for $\nabla_\theta v_\theta(t, x)$ with a computation time insensitive to $n$?*

The answer is affirmative. Precisely, our main contribution is designing the unbiased *generator gradient* estimator of $\nabla_\theta v_\theta(t, x)$ that requires only simulating $O(d^2)$ SDEs when the volatility parameters $\sigma_\theta$ do not depend on $\theta$ and $O(d^3)$ SDEs in the general setting, as summarized in Table 1.

We remark that in addition to pathwise differentiation, likelihood ratio-based estimators are also popular for sensitivity analysis in SDEs; see e.g. Yang and Kushner [26]. However, typically they are only applicable if $\sigma_\theta$ is independent of $\theta$ and under more restrictive jump structures. When applicable, likelihood ratio-based estimators could be appealing alternatives as they introduce a change of measure that represents the derivatives as a functional of the $d$-dimensional processes $X_\theta^x$. Nevertheless, these estimators typically have significantly higher variance.

Table 1: Comparison of the dimensions of SDEs needed to be simulated.

| Estimator | If the volatility depends on $\theta$ | |
|---|---|---|
| | Yes | No |
| Pathwise Differentiation | $d + d \cdot n$ | $d + d \cdot n$ |
| Generator Gradient | $d + d^2 + \frac{1}{2}d^3$ | $d + d^2$ |

Finally, we apply our estimator to linear-quadratic control problems and test its performance in optimizing neural-network-parameterized controls. As we increase the number of network parameters $n$, the results in Figure 1a and Table 2 highlight a substantial improvement in computational efficiency, as compared to the pathwise differentiation method, while still maintaining competitive variance levels. Furthermore, Figure 1a confirms that the computation time of our estimator is robust to increases in $n$, even in extremely high-dimensional scenarios with $n$ approaching $10^8$.

## 1.1 Literature Review

**Gradient Estimation:** Gradient estimation, particularly likelihood ratios and IPA methods, is crucial in sensitivity analysis. Foundational works in the late 20th century by Glynn [5, 4] and further adaptations to the SDE setting [26, 3] highlight these developments. IPA has evolved to apply stochastic flow techniques to SDEs, both with and without reflecting boundaries [23, 12, 16, 24, 14].

**Applications of Gradient Estimators:** Gradient estimators are widely used in stochastic control and reinforcement learning (RL) models. Policy gradient methods in discrete-time RL, including REINFORCE and deep policy gradient approaches, are notable applications [25, 13, 21]. Continuous-time RL have been explored using policy gradients in settings with continuous diffusion dynamics [8]. Jump diffusions are important models in financial engineering and stochastic control [17, 15, 9, 6]. Gradient estimators can also be used for optimizing these models. Neural SDE models are modern computational frameworks that model the dynamics of stochastic systems using a neural-network-parameterized SDE. Chen et al. [1], Tzen and Raginsky [23], Kidger [10] focus on the continuous case, while Jia and Benson [7] consider ODEs modulated by compound Poisson jumps. Efficient gradient estimators in high-dimensional settings are crucial for fitting these SDE models.

**Diffusion with Jumps and Stochastic Flow:** The main technical tools for this paper are SDEs with jumps and stochastic flows. Our references are Protter [19], Kunita [11], Øksendal and Sulem [18].

## 1.2 Remarks on Paper Organizations

The paper is structured as follows: Section 2 outlines the core concepts of our estimator in a zero-jump setting, focusing on intuitive understanding over technical detail. In Section 3, we introduce the SDE model with jumps and provide a set of sufficient conditions that rigorously support the earlier insights. While more general and complex assumptions exist that lead to similar conclusions, these are presented in Appendix A to align with the concise format of the conference proceedings. The paper concludes with Section 4, where we conduct numerical experiments on neural-network-parameterized linear-quadratic control problems, demonstrating the effectiveness of our methodology.

## 2 Key Methodological Insights

In this section, we motivate our proposed generator gradient estimator by first providing a non-rigorous derivation. We assume the SDE model (1.1) where the jumps $J_\theta \equiv 0$ and $\Theta \subset \mathbb{R}^n$ is a bounded open neighbourhood of the origin. W.l.o.g, we are interested in estimating the gradient at $\theta = 0 \in \Theta$ and $t = 0$; i.e. $\nabla_\theta v_0(0, x) = \nabla_\theta v_\theta(0, x)|_{\theta=0}$.

To simplify notation, we denote $X_\theta^x(t) := X_\theta^x(0, t)$ and $X_\theta^x(t-) := X_\theta^x(0, t-)$, and the function

$$a_{\theta,i,j}(t, x) := \frac{1}{2}\sum_{k=1}^{d'} \sigma_{\theta,i,k}(t, x)\sigma_{\theta,j,k}(t, x). \tag{2.1}$$

Also, for function $v_\theta(t, x)$, we use $\partial_i v_\theta(t, x)$ to denote the space derivative $\frac{\partial_i v}{\partial x}\big|_{\theta, t, x}$ and $\nabla$ the space gradient. Similarly, $\partial_{\theta_i}$ and $\nabla_\theta$ denotes the $\theta$ partials.

Under sufficient regularity conditions, by the Feynman-Kac formula, $v_\theta$ in (1.2) is the solution to the partial differential equation (PDE)

$$\partial_t v_\theta + \mathcal{L}_\theta v_\theta + \rho_\theta = 0, \quad v_\theta(T, \cdot) = g_\theta \tag{2.2}$$

for all $\theta \in \Theta$, where $\mathcal{L}_\theta$ is the *generator* of $X_\theta^x$ given by

$$\mathcal{L}_\theta f(t, x) := \sum_{i=1}^d \mu_{\theta, i}(t, x) \partial_i f(t, x) + \sum_{i,j=1}^d a_{\theta, i, j}(t, x) \partial_i \partial_j f(t, x)$$

for $f$ that is twice differentiable in $x$. Assuming enough smoothness, we formally differentiate the PDE (2.2) w.r.t. $\theta_i$ and then set $\theta = 0$ to obtain

$$\partial_t \partial_{\theta_i} v_0 + \mathcal{L}_0 \partial_{\theta_i} v_0 + (\partial_{\theta_i} \mathcal{L}_0 v_0 + \partial_{\theta_i} \rho_0) = 0, \quad \partial_{\theta_i} v_0(T, \cdot) = \partial_{\theta_i} g_0. \tag{2.3}$$

Here, the operator $\partial_{\theta_i} \mathcal{L}_0$ is defined as

$$\partial_{\theta_i} \mathcal{L}_0 f(t, x) := \sum_{j=1}^d \partial_{\theta_i} \mu_{0, j}(t, x) \partial_j f(t, x) + \sum_{j,l=1}^d \partial_{\theta_i} a_{0, j, l}(t, x) \partial_j \partial_l f(t, x). \tag{2.4}$$

Interpreted as the derivative of $\mathcal{L}_\theta$ w.r.t. $\theta$ at 0, this inspires the name "generator gradient" method.

Next, define $u_0 = \partial_{\theta_i} v_0$. Treating $\partial_{\theta_i} \mathcal{L}_0 v_0$ as fixed, we observe that $u_0$ solves the PDE (2.3) which is of the form (2.2). Hence, applying the Feynman-Kac formula again to $\partial_{\theta_i} v_0(0, x) = u_0(0, x)$ yields the following expectation representation

$$\partial_{\theta_i} v_0(0, x) = E\left[ \int_0^T \partial_{\theta_i} \mathcal{L}_0 v_0(t, X_0^x(t)) + \partial_{\theta_i} \rho_0(t, X_0^x(t)) dt + \partial_{\theta_i} g_0(X_0^x(T)) \right]. \tag{2.5}$$

Note that the expression inside the expectation contains only space derivatives (due to $\partial_{\theta_i} \mathcal{L}_0$) of the value function $v_0$ but not the $\theta$ derivatives. In particular, if we can estimate the gradient $\nabla v_0(t, x)$ and the Hessian matrix $H[v_0](t, x) := \{\partial_i \partial_j v_0(t, x) : 1 \le i, j \le d\}$ efficiently, then the representation in (2.5) will lead to a natural estimator of $\partial_{\theta_i} v_0(0, x)$.

To estimate $\nabla v_0(t, x)$ and $H[v_0](t, x)$, we employ the pathwise differentiation estimator from (1.3). Specifically, under enough regularity conditions, we can interchange the derivatives and integration

$$\nabla v_0(t, x)^\top = EZ(t, x)^\top := E\left[ \int_t^T \nabla \rho_0^\top \nabla X_0^x(t, r) dr + \nabla g_0^\top \nabla X_0^x(t, T) \right],$$

$$H[v_0](t, x) = EH(t, x) := E\left[ \nabla X_0^x(t, T)^\top H[g_0] \nabla X_0^x(t, T) + \langle \nabla g_0, H[X_{0, \cdot}^x](t, T) \rangle \right] \tag{2.6}$$

$$+ E\left[ \int_t^T \nabla X_0^x(t, r)^\top H[\rho_0] \nabla X_0^x(t, r) + \langle \nabla \rho_0, H[X_{0, \cdot}^x](t, r) \rangle dr \right].$$

Here, we write $\nabla X_0^x := \{\partial_a X_{0, i}^x : i, a = 1, \ldots d\}$ and $H[X_0^x] := \{\partial_b \partial_a X_{0, i}^x : i, a, b = 1, \ldots d\}$. The notation $\langle \nabla h, H[X_{0, \cdot}^x] \rangle := \sum_{a=1}^d \partial_a h H[X_{0, a}^x] \in \mathbb{R}^{d \times d}$ for $h = \rho_0, g_0$. The dependence of $\rho_0, g_0$ on time and the state process is hidden.

We estimate these expectations by simulating the SDEs for $\{X_0^x, \nabla X_0^x, H[X_0^x]\}$ given by (1.1) and

$$\partial_a X_{0, i}^x = \delta_{i, a} + \int_t^s \sum_{l=1}^d \partial_l \mu_{0, i} \partial_a X_{0, l}^x dr + \int_t^s \sum_{l=1}^d \sum_{k=1}^{d'} \partial_l \sigma_{0, i, k} \partial_a X_{0, l}^x dB_k(r)$$

$$\partial_b \partial_a X_{0, i}^x = \int_t^s \sum_{l=1}^d \left[ \partial_l \mu_{0, i} \partial_b \partial_a X_{0, l}^x + \sum_{m=1}^d \partial_m \partial_l \mu_{0, i} \partial_a X_{0, l}^x \partial_b X_{0, m}^x \right] dr \tag{2.7}$$

$$+ \int_t^s \sum_{k=1}^{d'} \sum_{l=1}^d \left[ \partial_l \sigma_{0, i, k} \partial_b \partial_a X_{0, l}^x + \sum_{m=1}^d \partial_m \partial_l \sigma_{0, i, k} \partial_a X_{0, l}^x \partial_b X_{0, m}^x \right] dB_k(r)$$

where the dependence of the coefficients on $r$, $X_0^x(t, r-)$, and $z$, as well as the dependence of $X_0^x, \partial_a X_0^x, \partial_a \partial_b X_0^x$ on $(t,s), (t, r-)$ are suppressed.

The dimension of these SDEs is $d + d^2 + \frac{1}{2}d^3$, where the $\frac{1}{2}$ comes from the Hessian being symmetric. Moreover, when the volatility $\sigma$ is independent of $\theta$, our method only necessitates estimating $\nabla v_0$. This reduction leads to simulating the SDEs for $\{X_0^x, \nabla X_0^x\}$ of dimension only $d + d^2$.

Assuming sufficient integrability, the unbiasedness of $Z$ implies

$$E\left[\int_0^T \partial_{\theta_k} \mu_0^\top Z(t, X_0^x(0, t))dt\right] = E\left[\int_0^T \partial_{\theta_k} \mu_0^\top \nabla v_0(t, X_0^x(0, t))dt\right] \qquad (2.8)$$

which we will elaborate upon in (A.1). The same holds for the $H(t, x)$ process as well. Therefore, we can replace the derivatives $\nabla v_0$ with $Z$ and $H[v_0]$ with $H$ in (2.5) without changing the expectation.

Also note that producing a sample of $Z(t, x)$ requires simulating the solution to SDEs (1.1) and (2.7) within time $[t, T]$ starting from $x, I, 0$. So, it is not very efficient to compute $Z(t, X_0^x(0, t))$ for every $t$; a similar issue exists for $H$ as well. This can be addressed by randomizing the integral.

With these considerations, we proceed to define the generator gradient estimator. First, let $\nabla_\theta L_0 V_0(t, x)$ be defined by replacing $\partial_i v(t, x)$ with $Z_i(t, x)$ and $\partial_j \partial_i v$ with $H_{i,j}(t, x)$ in the definition (2.4) of $\partial_{\theta_i} \mathcal{L}_0 v_0(t, x)$. Then, define the generator gradient estimator as

$$D(x) := T\nabla_\theta L_0 V_0(\tau, X_0^x(0, \tau)) + \int_0^T \nabla_\theta \rho_0(t, X_0^x(t))dt + \nabla_\theta g_0(X_0^x(T)). \qquad (2.9)$$

where $\tau \sim \text{Unif}[0, T]$ is sampled independently. We can also randomize the integral of $\nabla_\theta \rho_0(t, X_\theta^x(t))$ if the gradient is hard to compute. With the derivation in (2.8), it is easy to see that $ED(x) = \nabla_\theta v_0(0, x)$ is unbiased.

In summary, due to the observation in (2.5), we are able to "move" the estimation of $\nabla_\theta v_0$ onto that of $\nabla v_0$ and $H[v_0]$. This results in a significant reduction in the dimension of the SDEs we need to simulate, underlying the remarkable efficiency of our methodology, especially when the dimension $n$ of $\theta$ significantly exceeds $d$.

## 3  Jump Diffusions and the Generator Gradient Estimator

In this section, we rigorously formulate a jump diffusion process driven by an SDE. We extend the generator gradient estimator to this context by first rigorously establishing an expectation representation of the derivative as in (2.5). Then, we also validate the representation (2.6) using the jump version of (2.7). These lead to our generator gradient estimator in the jump diffusion context. To improve the clarity of the paper (at a cost of generalizability), we will state a set of sufficient assumptions that are easy to verify. However, we will state and prove our theorems using a set of more general assumptions in the Appendix A.

We consider jump diffusions on the canonical probability space of càdlàg functions $[0, T] \to \mathbb{R}^d$ generated by SDEs of the form (1.1) where the jump term is given by

$$\begin{aligned} X_{\theta,i}^x(t, s) = x_i &+ \int_t^s \mu_{\theta,i}(r, X_\theta^x(t, r))dr + \int_t^s \sum_{k=1}^{d'} \sigma_{\theta,i,k}(r, X_\theta^x(t, r-))dB_k(r) \\ &+ \int_t^s \int_{\mathbb{R}_0^{d'}} \chi_{\theta,i}(t, X_\theta^x(s, r-), z)d\widetilde{N}(dr, dz). \end{aligned} \qquad (3.1)$$

In this expression, $B$ is a standard Brownian motion in $\mathbb{R}^{d'}$; $\widetilde{N}$ is a compensated Poisson random measure with intensity measure $dt \times \nu(dz)$ with $\nu$ a Lévy measure on $(\mathbb{R}_0^{d'} := \mathbb{R}^{d'} \setminus \{0\}, \mathcal{B}(\mathbb{R}_0^{d'}))$, i.e. $\int_{\mathbb{R}_0^{d'}} 1 \wedge |z|^2 \nu(dz) < \infty$; the $-r$ notation in $X_\theta^x(t, r-)$ denotes the left limit; and the stochastic integrations are Itô integrals. Here, for a vector/matrix/tensor $v \in \mathbb{R}^{d_1 \times d_2 \times d_3}$, we denote $|v|^2 := \sum_{i,j,k} |v_{i,j,k}|^2$. We further define $\gamma(z) = |z| \wedge 1$ and $\mu(dz) = \gamma(z)^2 \nu(dz)$. Then $\mu$ is a finite measure on $(\mathbb{R}_0^{d'}, \mathcal{B}(\mathbb{R}_0^{d'}))$. Also, since we are interested in the gradient at $\theta = 0$, we can assume w.l.o.g. that $\Theta$ is a bounded open neighbourhood of 0.

The generator of this system of SDEs is $\mathcal{L}_\theta := \mathcal{L}_\theta^C + \mathcal{L}_\theta^J$, where

$$\mathcal{L}_\theta^C f(t,x) = \sum_{i=1}^d \mu_{\theta,i}(t,x)\partial_i f(t,x) + \sum_{i,j=1}^d a_{\theta,i,j}(t,x)\partial_i\partial_j f(t,x)$$

$$\mathcal{L}_\theta^J f(t,x) = \int_{\mathbb{R}_0^{d'}} \left[ f(t, x + \chi_\theta(t,x,z)) - f(t,x) - \sum_{i=1}^d \chi_{\theta,i}(t,x,z)\partial_i f(t,x) \right] \nu(dz).$$

(3.2)

for $f \in C^{1,2}([0,T],\mathbb{R}^d)$. We remark that for open subsets $\mathbb{W}, \mathbb{X}$, the space $C^{i,j,k}([0,T],\mathbb{W},\mathbb{X})$ represents the set of functions $f$ on $[0,T] \times \mathbb{W} \times \mathbb{X}$ that has continuous mixed partial derivatives $\partial_t^a \partial_w^b \partial_x^c f$ on $(0,T) \times \mathbb{W} \times \mathbb{X}$ for every $a \le i, b \le j, c \le k$. Moreover, these mixed partial derivatives have continuous extensions on $[0,T] \times \mathbb{W} \times \mathbb{X}$.

### 3.1 Probabilistic Representation of the Gradient

In this section, we rigorously establish the probabilistic representation of the gradient $\nabla_\theta v_0(0,x)$ as outlined in equation (2.5). Our approach leverages the continuous dependence of $\theta \to X_\theta^x$ of the solutions to (3.1) in a neighbourhood of $0$, given sufficient regularity conditions. This behavior extends the properties associated with stochastic flows, as explained in the work by Kunita [11].

Recall that $\Theta$ is a bounded neighbourhood of $0 \in \mathbb{R}^n$. To clarify the assumptions, we enlarge $\Theta$ and consider $\Theta_\epsilon = \{\theta + v : \theta \in \Theta, v \in B^n(0,\epsilon)\}$ where $B^n(0,\epsilon)$ is the open ball in $\mathbb{R}^n$ at $0$ of radius $\epsilon$.

**Assumption 1.** *For some $\epsilon > 0$, the following regularity conditions hold*

1. *The mappings $(s,\theta,x) \to \mu_\theta(s,x), \sigma_\theta(s,x), \rho_\theta(s,x), g_\theta(s,x)$ are $C^{0,1,1}([0,T],\Theta_\epsilon,\mathbb{R}^d)$. For each $z \in \mathbb{R}_0^{d'}$, $(s,\theta,x) \to \chi_\theta(s,x,z)/\gamma(z)$ is $C^{0,1,1}([0,T],\Theta_\epsilon,\mathbb{R}^d)$. Moreover, $|\chi_\theta(s,0,z)/\gamma(z)|$ is uniformly bounded in $s \in [0,T]$ and $z \in \mathbb{R}_0^{d'}$.*

2. *The spacial derivatives $|\nabla\mu_\theta|$, $|\nabla\sigma_\theta|$, and $|\nabla\chi_\theta|$ are uniformly bounded. The $\theta$ derivatives satisfy linear growth*

$$|\nabla_\theta\mu_\theta(s,x)| + |\nabla_\theta\sigma_\theta(s,x)| + \left|\frac{\nabla_\theta\chi_\theta(s,x,z)}{\gamma(z)}\right| \le \ell(|x|+1)$$

*for all $s \in [0,T]$, $x \in \mathbb{R}^d$, $z \in \mathbb{R}^{d'}$, and $\theta \in \Theta$.*

3. *The $\theta$ derivatives of the rewards satisfy polynomial growth: for some $m \ge 1$,*

$$|\nabla_\theta\rho_\theta(s,x)| + |\nabla_\theta g_\theta(x)| \le \ell(|x|+1)^m$$

*for all $s \in [0,T]$, $x \in \mathbb{R}^d$, and $\theta \in \Theta$.*

*Remark.* Requirement 1 implies that for each fixed $x$, the $\theta$ derivatives of the coefficients are uniformly bounded in $[0,T] \times \Theta$, as $\Theta$ is assumed to be bounded. So, the seemingly strong requirements of the $\theta$ derivative satisfying the growth condition in items 2 and 3 are not very restrictive. The boundedness of $\chi_\theta(s,x,z)/\gamma(z)$ in $z$ is relaxed in Assumption 5 in the appendix, allowing unbounded jumps. The strong condition is the uniform boundedness of $|\nabla\mu_\theta|$, $|\nabla\sigma_\theta|$, and $|\nabla\chi_\theta|$. However, this is typically necessary for the existence and uniqueness of strong solutions to the SDE (3.1).

**Assumption 2.** *Assume that $\{v_\theta \in C^{1,2}([0,T],\mathbb{R}^d) : \theta \in \Theta\}$ are classical solutions to the partial-integro-differential equations (PIDE)*

$$\partial_t v_\theta + \mathcal{L}_\theta v_\theta + \rho_\theta = 0, \qquad v_\theta(T,\cdot) = g_\theta$$

*where $\mathcal{L}_\theta = \mathcal{L}_\theta^C + \mathcal{L}_\theta^J$ are defined in (3.2). Moreover, $v_\theta$ and its space derivatives satisfy polynomial growth: for each $\theta \in \Theta$, there exists $0 < c_\theta < \infty$ and $m \ge 1$ s.t.*

$$\sup_{x\in\mathbb{R}^d, t\in[0,T]} \frac{|v_\theta(t,x)|}{(|x|+1)^m} \le c_\theta, \qquad \sup_{x\in\mathbb{R}^d, t\in[0,T]} \frac{|\nabla v_\theta(t,x)|}{(|x|+1)^m} \le c_\theta, \qquad \sup_{x\in\mathbb{R}^d, t\in[0,T]} \frac{|H[v_\theta](t,x)|}{(|x|+1)^m} \le c_\theta.$$

*Remark.* By classical solution, we mean that $v_\theta$ satisfies $\partial_t v_\theta + \mathcal{L}_\theta v_\theta + \rho_\theta = 0$ on $(0,T) \times \mathbb{R}^d$ with its continuous extensions of satisfying $v_\theta(T,\cdot) = g_\theta$. This is possible, for example, in settings with $C^2$ terminal rewards. Note that is a stronger requirement compared to the definition in Evans [2].

As we have motivated in Section 2, Assumption 2 follows from a generalized version of the Feynman-Kac formula, under additional technical assumptions. Moreover, the growth of $v_\theta$ and its space derivatives can be derived from assumptions on the growth of the rewards. However, in order to not obscure the main message of the paper and to streamline the proof, we directly assume these properties. We refer interested readers to Kunita [11, Chapter 4] where stochastic flow techniques similar to the proofs in the paper are employed to establish the PIDE and validate the growth rates.

**Theorem 1** (Probabilistic Representation of the Gradient). *If Assumptions 1 and 2 are in force, then $\theta \to v_\theta(0, x)$ is differentiable at $0$. Moreover, the gradient*

$$\nabla_\theta v_0(0, x) = E\left[\int_0^T \nabla_\theta \mathcal{L}_0 v_0(s, X_0^x(s)) + \nabla_\theta \rho_\theta(X_0^x(s))ds + \nabla_\theta g_\theta(X_0^x(T))\right],$$

*where $\nabla_\theta \mathcal{L}_0 := \nabla_\theta \mathcal{L}_0^C + \nabla_\theta \mathcal{L}_0^J$ s.t. for $f(t, x) \in C^{1,2}$,*

$$\nabla_\theta \mathcal{L}_\theta^C f(t, x) = \sum_{i=1}^d \nabla_\theta \mu_{\theta,i}(t, x)\partial_i f(t, x) + \sum_{i,j=1}^d \nabla_\theta a_{\theta,i,j}(t, x)\partial_i\partial_j f(t, x), \tag{3.3}$$

$$\nabla_\theta \mathcal{L}_\theta^J f(t, x) = \int_{\mathbb{R}_0^{d'}} \left[\sum_{i=1}^d \nabla_\theta \chi_{\theta,i}(t, x, z) \left(\partial_i f(t, x + \chi_\theta(t, x, z)) - \partial_i f(t, x)\right)\right] \nu(dz). \tag{3.4}$$

In Theorem 1, we have successfully established an expectation representation of the gradient $\nabla_\theta v_0(0, x)$ of the form (2.5). This naturally leads to the consideration of using Monte Carlo to estimate $\nabla_\theta v_0(0, x)$. However, one observes that the representation in Theorem 1 involves the space derivatives $\partial_i v_0(t, x)$ and $\partial_i\partial_j v_0(t, x)$, which are usually hard to compute exactly.

In the next section section, following the heuristics in (2.6) we establish conditions on the model primitives so that the space derivatives $\partial_i v_0(t, x)$ and $\partial_i\partial_j v_0(t, x)$ admit probabilistic representations as expectations of random processes $\{X_0^x, \nabla X_0^x, H[X_0^x]\}$ that can be easily simulated.

## 3.2 Probabilistic Representation of the Space Derivatives

We proceed with introducing assumptions that guarantee Theorem 2, providing representations of $\partial_i v_0(t, x)$ and $\partial_i\partial_j v_0(t, x)$ as illustrated in (2.6). To achieve this, we first need to ensure that the derivative of the mapping $x \to X_0^x$ is well defined. This is formally established in Proposition A.1.

**Assumption 3.** *For each $z \in \mathbb{R}_0^{d'}$, the SDE coefficients $(s, x) \to (\mu_0(s, x), \sigma_0(s, x), \chi_0(s, x, z))$ are $C^{0,2}([0, T], \mathbb{R}^d)$. For each $i, j = 1, \ldots, d$, the coefficients and derivatives, seen as functions $(s, x) \to (\alpha(s, x), \beta(s, x), \zeta(s, x, \cdot))$ where $(\alpha, \beta, \zeta) = (\mu_0, \sigma_0, \chi_0/\gamma)$, $(\partial_i \mu_0, \partial_i \sigma_0, \partial_i \chi_0/\gamma)$, and $(\partial_j \partial_i \mu_0, \partial_j \partial_i \sigma_0, \partial_j \partial_i \chi_0/\gamma)$ are uniformly Lipschitz; i.e. there exists $0 \le \ell < \infty$ s.t. for all $s \in [0, T], z \in \mathbb{R}^{d'}$*

$$|\alpha(s, x) - \alpha(s, x')| + |\beta(s, x) - \beta(s, x')| + |\zeta(s, x, z) - \zeta(s, x', z)| \le \ell |x - x'|.$$

*Moreover, $|\zeta(s, 0, z)|$ is uniformly bounded for $s \in [0, T]$ and $z \in \mathbb{R}_0^{d'}$.*

In view of this assumption, we consider the following SDEs, as jump versions of (2.7), for which the strong solutions should be the space derivatives of $X_0^x$. Again, the dependence of the coefficients on $r$, $X_0^x(t, r-)$, and $z$, as well as the dependence of $X_0^x, \partial_a X_0^x, \partial_a \partial_b X_0^x$ on $(t, s), (t, r-)$ has been

suppressed.

$$\partial_a X_{0,i}^x = \delta_{i,a} + \int_t^s \sum_{l=1}^d \partial_l \mu_{0,i} \partial_a X_{0,l}^x dr + \int_t^s \sum_{l=1}^d \sum_{k=1}^{d'} \partial_l \sigma_{0,i,k} \partial_a X_{0,l}^x dB_k(r)$$

$$+ \int_t^s \sum_{l=1}^d \partial_l \chi_{0,i} \partial_a X_{0,l}^x d\widetilde{N}(dr, dz)$$

$$\partial_b \partial_a X_{0,i}^x = \int_t^s \sum_{l=1}^d \left[ \partial_l \mu_{0,i} \partial_b \partial_a X_{0,l}^x + \sum_{m=1}^d \partial_m \partial_l \mu_{0,i} \partial_a X_{0,l}^x \partial_b X_{0,m}^x \right] \tag{3.5}$$

$$+ \int_t^s \sum_{k=1}^{d'} \sum_{l=1}^d \left[ \partial_l \sigma_{0,i,k} \partial_b \partial_a X_{0,l}^x + \sum_{m=1}^d \partial_m \partial_l \sigma_{0,i,k} \partial_a X_{0,l}^x \partial_b X_{0,m}^x \right] dB_k(r)$$

$$+ \int_t^s \sum_{l=1}^d \left[ \partial_l \chi_{0,i} \partial_b \partial_a X_{0,l}^x + \sum_{m=1}^d \partial_m \partial_l \chi_{0,i} \partial_a X_{0,l}^x \partial_b X_{0,m}^x \right] d\widetilde{N}(dr, dz).$$

As we will show in Proposition A.1, under Assumption 3 the process $X_0^x(t, s)$ has a version that is twice continuously differentiable in $x$ for every $0 \le t < s \le T$. The processes $\{\nabla X_0^x, H[X_0^x]\}$, as defined in (3.5), will then correspond to the derivatives. Moreover, these processes, as well as $X_0^x$, will possess desirable integrability properties.

To guarantee sufficient integrability and to provide a variance bound for our estimator, we also need to assume growth conditions on the rewards.

**Assumption 4.** *Assume that the mapping* $x \to \rho_0(t, x), g_0(x)$ *is* $C^2$ *for all* $t \in [0, T]$. *Moreover, for* $h(t, x) = \rho_0(t, x)$ *and* $g_0(x)$ *there exists* $c_h$ *s.t.*

$$\sup_{x \in \mathbb{R}^d, t \in [0,T]} \frac{|h(t, x)|}{(|x| + 1)^m} \le c_h, \qquad \sup_{x \in \mathbb{R}^d, t \in [0,T]} \frac{|\nabla h(t, x)|}{(|x| + 1)^m} \le c_h, \qquad \sup_{x \in \mathbb{R}^d, t \in [0,T]} \frac{|H[h](t, x)|}{(|x| + 1)^m} \le c_h.$$

With these assumptions, we validate the representations in (2.6) using the following theorem.

**Theorem 2** (Probabilistic Representation of the Space Derivatives)**.** *Under Assumptions 3 and 4, the representations in* (2.6) *hold with the jump version of* $\{X_0^x, \nabla X_0^x, H[X_0^x]\}$ *in* (3.1) *and* (3.5)*.*

### 3.3 The Generator Gradient Estimator

With Theorems 1 and 2, we construct our generator gradient estimator and show that it is unbiased with a variance that grows polynomially in $x$. Recall the estimators $Z(t, x)$ and $H(t, x)$ in (2.6).

By Theorem 2 and the integrability in Proposition A.1 under Assumption 3, the equality (2.8) holds. Then, following the notation in (2.9), we define

$$\nabla_\theta L_0 V_0(t, x) := \nabla_\theta L_0^C V_0(t, x) + \nabla_\theta L_0^J V_0(t, x)$$

where $\nabla_\theta L_0^C V_0(t, x)$ and $\nabla_\theta L_0^J V_0(t, x)$ are defined by replacing $\partial_i v(t, x)$ with $Z_i(t, x)$ and $\partial_j \partial_i v$ with $H_{i,j}(t, x)$ in (3.3) and (3.4), respectively. Then, our estimator $D(x)$ is given by (2.9).

**Theorem 3.** *Suppose Assumptions 1-4 are in force. Then, the generator gradient estimator* $D(x)$ *is unbiased; i.e.* $ED(x) = \nabla_\theta v_0(0, x)$. *Moreover, the variance* $\mathrm{Var}(D(x)) \le C(|x| + 1)^{2m+4}$ *has at most polynomial growth in* $x$, *where the constant* $C$ *can be dependent on other parameters of the problem but not* $x$.

*Remark.* The $m$ signifies the growth rate of the rewards and their derivatives. The extra additive factor 2 in the variance is from the growth of the $\theta$ derivative of $a_0$, the volatility squared.

## 4 Example: Linear System with Quadratic Loss

In this section, we illustrate some analytical properties and the effectiveness of our estimator by considering a linear quadratic control problem.

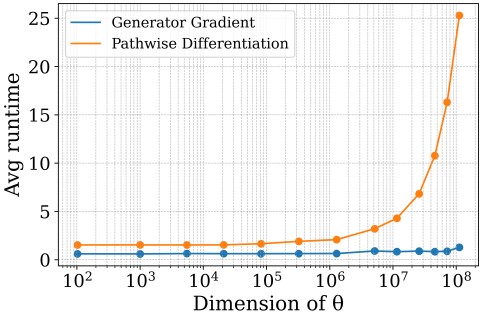 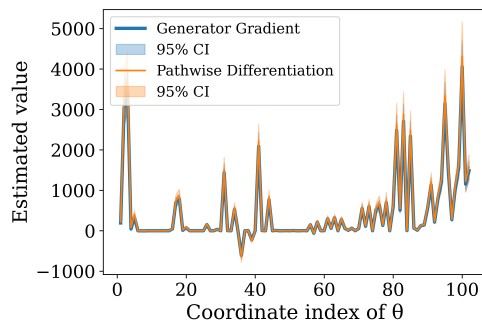

(a) Average runtime for different $n$.

(b) Estimator statistics with $n = 102$.

Figure 1: Comparisons of 100-sample estimation statistics and averaged runtime.

Let $X \in \mathbb{R}^d$ be the controlled process, given by the solution to the SDE

$$X^x(t) = x + \int_0^t AX^x(s) + BU(t)ds + \int_0^t CdB(s),$$

where $B(t) \in \mathbb{R}^{d'}$ is a standard Brownian motion, $U(t) \in \mathbb{R}^m$ is the control process that is adapted to the filtration generated by $X$, $A \in \mathbb{R}^{d \times d}, B \in \mathbb{R}^{d \times m}, C \in \mathbb{R}^{d \times d'}$ are non-random matrices. The objective is to choose an admissible control $U(t)$ that minimizes the quadratic loss

$$E\left[\int_0^T X^x(t)^\top Q X^x(t) + U(t)^\top R U(t)dt + X^x(T)^\top Q_T X^x(T)\right]$$

where $Q, Q_T \in \mathbb{R}^{d \times d}$ and $R \in \mathbb{R}^{m \times m}$ are non-random matrices.

In various applications of interests, the admissible control $U(t)$ is a parameterized function of time and state $U(t) = u_\theta(t, X^x_\theta(t))$ where the state process under control $u_\theta$ is denoted by $X^x_\theta$. The dimension of $\theta$ could potentially be very high—e.g. when $u_\theta$ is a neural network. To achieve an optimized loss in this over-parameterized setting, one common approach is to run gradient descent. Hence, an efficient gradient estimator that scales well with the dimension $n$ of $\theta$ is highly desirable.

We compare the performance of the proposed generator gradient estimator and the pathwise differentiation estimator. In this context, these estimators take the following form. The detailed derivations are presented in Appendix F.1.

**The Generator Gradient Estimator:** In this setting, our generator gradient estimator in (2.9) is

$$D_i(x) = T\partial_{\theta_i} u_\theta(\tau, X^x_\theta(\tau))^\top B^\top Z(\tau, X^x_\theta(\tau)) + Tu_\theta(\tau, X^x_\theta(\tau))^\top (R + R^\top)\partial_{\theta_i} u_\theta(\tau, X^x_\theta(\tau))$$

where the definition of $Z$ follows from (2.6), and is given by (F.1) in Appendix F.1. As explained in (2.9), we also randomize the integral corresponding to the gradient of the reward rate $\nabla_\theta \rho_0$.

**The Pathwise Differentiation Estimator:** From (1.3), we find the following IPA estimator that randomizes the time integral

$$\widetilde{D}_i(x) = Tu_\theta(\tau, X^x_\theta(\tau))(R + R^\top)\nabla u_\theta(\tau, X^x_\theta(\tau))\partial_{\theta_i} X^x_\theta(\tau) + TX^x_\theta(\tau)^\top (Q + Q^\top)\partial_{\theta_i} X^x_\theta(\tau)$$
$$+ Tu_\theta(\tau, X^x_\theta(\tau))^\top (R + R^\top)\partial_{\theta_i} u_\theta(\tau, X^x_\theta(\tau)) + X^x_\theta(T)^\top (Q_T + Q_T^\top)\partial_{\theta_i} X^x_\theta(T).$$

Here, the pathwise derivatives $\partial_{\theta_i} X^x_\theta(t)$ is the solution to (F.3).

We deploy these estimators in an environment where the state variable $x \in \mathbb{R}^4$ represents the x-y positions and velocities of a point mass on a 2D plane. The controller applies a force to this mass. The cost function is designed to encourage the controller to swiftly move the point mass to the origin with minimal force. The force is state-time-dependent and parameterized through a 4-layer fully connected neural network with variable width. All computation times are recorded from a Tesla V100 GPU. Further details about the setup of our numerical experiments can be found in Appendix F.2.

In Figure 1a, we present a comparison of the average runtime for computing a single sample of the generator gradient and the pathwise differentiation estimators $D(x), \widetilde{D}(x) \in \mathbb{R}^n$, across increasing values of $n$ the dimension of $\theta$. Our findings indicate that the generator gradient estimator not only outperforms the widely used pathwise differentiation method across all tested values of $n$ but also surpasses it by more than an order of magnitude for larger values of $n$. Additionally, the computation time for our estimator shows remarkable stability with respect to increases in $n$, displaying only a slight uptrend when $n \gtrsim 10^7$.

Figure 1b confirms that, at $n = 102$, the estimated values by the two estimators are very similar with high confidence. This confirms that our estimator is consistently estimating the gradient $\nabla_\theta v_\theta(0, x)$.

Table 2: 400-sample standard error (SE) comparison between generator gradient (GG) and pathwise differentiation (PD) estimators.

| $n$ (dimension of $\theta$) | 102 | 1002 | 5502 | 21002 | 3.24e5 | 1.29e6 | 5.14e6 | 1.15e7 |
|---|---|---|---|---|---|---|---|---|
| Avg SE of GG | 5.253 | 5.785 | 3.533 | 1.205 | 0.965 | 0.729 | 0.600 | 0.407 |
| Avg SE of PD | 6.424 | 5.710 | 4.453 | 1.191 | 1.110 | 0.935 | 0.786 | 0.466 |
| Avg SE ratios | 0.971 | 0.932 | 0.946 | 0.903 | 0.902 | 0.914 | 0.926 | 0.961 |

Finally, Table 2 presents the standard errors (SE) (F.4) from 400 replications of both estimators, averaged over the gradient coordinates. It also displayed the averaged ratios of the standard errors (F.5). We observe averaged SE ratios that are consistently less than 1 for all $n$, suggesting that our generator gradient estimator not only provides significantly faster computations as shown in Figure 1a but also achieves lower estimation variances. Further analysis of the SEs for each gradient coordinate is conducted and displayed in Figure 2 in Appendix F.2, highlighting similar histogram shapes and observable reduction in large values of SEs of our estimator.

## 5  Concluding Remarks

The theoretical results in this paper have the limitation of requiring second-order continuous differentiability and uniform boundedness of the space derivatives of the parameters of the underlying jump diffusion. These strong conditions, which are standard in the literature of stochastic flows (cf. [19, 11]) to guarantee global existence and uniqueness of the derivative processes in (3.5), are necessary to achieve the generality of the results presented in this paper.

However, our generator gradient estimator often works even when coefficients are not continuously differentiable. This is true if the generator and rewards gradients are defined almost everywhere, and the derivative processes in (3.5), with almost everywhere derivatives of the SDE parameters, exist for every $t \in [0, T]$ and satisfy some integrability conditions. Examples include neural networks parameterized stochastic control with ReLU activation functions, heavy-traffic limits of controlled multi-server queues, and the Cox–Ingersoll–Ross (CIR) model. For these models, the existence and integrability of the derivative processes can be checked on a case-by-case basis, allowing the consistency and unbiasedness of the generator gradient estimator to be established. We confirm this by numerically investigating the CIR process and an SDE with ReLU drift in Appendix G.

### Acknowledgments and Disclosure of Funding

The material in this paper is based upon work supported by the Air Force Office of Scientific Research under award number FA9550-20-1-0397. Additional support is gratefully acknowledged from NSF 2118199, 2229012, 2312204, and ONR 13983111.

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

# Appendices

## A    Generalizations of the Assumptions

### A.1    Probabilistic Representation of the Gradient

In this section, we develop a generalized version of Theorem 1, weakening Assumptions 1 and 2. In particular, we allow discontinuities in time of the SDE coefficients. This flexibility is especially relevant in applications in data-driven decision-making environments where non-homogeneous SDE models with estimated drift, volatility, and jump parameters could be piece-wise constant. Moreover, we also relax the differentiability of the coefficients in the space variable to Lipschitz continuity. We will state the new set of assumptions, and establish a generalized version of Theorem 1 as in Theorem 1'

We proceed by presenting a critical theorem, along with the necessary assumptions, that forms the foundation of our probabilistic representation in Theorem 1'.

**Assumption 5.** *Assume that for each $\theta$ the coefficients $\mu_\theta(\cdot, \cdot)$, $\sigma_\theta(\cdot, \cdot)$, and $\chi_\theta(\cdot, \cdot, \cdot)$ are jointly Borel measurable. Moreover, assume the following holds true:*

*1. At $x = 0$, the coefficients are bounded: for all $p \geq 2$,*

$$\sup_{\theta \in \Theta, s \in [0,T]} \left[ |\mu_\theta(s, 0)| + |\sigma_\theta(s, 0)| + \int_{\mathbb{R}_0^{d'}} \left| \frac{\chi_\theta(s, 0, z)}{\gamma(z)} \right|^p \mu(dz) \right] < \infty$$

*2. The coefficients are uniform Lipschitz in $x$, uniformly in $s, \theta$ in the following sense: there exists constants $c$ and $\{c_p : p \geq 2\}$ s.t.*

$$|\mu_\theta(s, x) - \mu_\theta(s, x')| \leq c\, |x - x'|, \quad |\sigma_\theta(s, x) - \sigma_\theta(s, x')| \leq c\, |x - x'|,$$

*and for any $p \geq 2$*

$$\left( \int_{\mathbb{R}_0^{d'}} \left| \frac{\chi_\theta(s, x, z)}{\gamma(z)} - \frac{\chi_\theta(s, x', z)}{\gamma(z)} \right|^p \mu(dz) \right)^{\frac{1}{p}} \leq c_p |x - x'|$$

*for all $s \in [0, T]$, $x, x' \in \mathbb{R}^d$, $\theta \in \Theta$.*

*3. The coefficients are weakly Lipschitz in $\theta$ the following sense: for each $p \geq 2$, there exists a time-dependent positive field $\left\{ \kappa_{\theta, \theta'}^p(s) \in \mathbb{R}_{>0} : s \in [0, T], \theta, \theta' \in \Theta \right\}$ s.t. for some constant $\ell_p$,*

$$\left( \int_0^T \kappa_{\theta, \theta'}^p(s) ds \right)^{\frac{1}{p}} \leq \ell_p\, |\theta - \theta'|$$

*for all $\theta, \theta' \in \Theta$, and the coefficients satisfy*

$$|\mu_\theta(s, x) - \mu_{\theta'}(s, x)|^p \leq \kappa_{\theta, \theta'}^p(s)(|x| + 1)^p, \quad |\sigma_\theta(s, x) - \sigma_{\theta'}(s, x)|^p \leq \kappa_{\theta, \theta'}^p(s)(|x| + 1)^p,$$

*and*

$$\int_{\mathbb{R}_0^{d'}} \left| \frac{\chi_\theta(s, x, z)}{\gamma(z)} - \frac{\chi_{\theta'}(s, x, z)}{\gamma(z)} \right|^p \mu(dz) \leq \kappa_{\theta, \theta'}^p(s)(|x| + 1)^p$$

*for all $s \in [0, T]$, $\theta, \theta' \in \Theta$, $x \in \mathbb{R}^d$.*

**Theorem K** (Theorem 3.3.1 of Kunita [11])**.** *If Assumption 5 is in force, then the family of solutions $X^x := \{ X_\theta^x(t) : t \in [0, T], \theta \in \Theta \}$ has a version $\hat{X}^x$ (i.e. $\left\{ \exists t \in [0, T], \theta \in \Theta : X_\theta^x(t) \neq \hat{X}_\theta^x(t) \right\} \subset N$ with $P(N) = 0$) that is $\mathcal{B}([0, T]) \times \mathcal{B}(\Theta) \times \mathcal{F}$ measurable. Moreover, w.p.1, for each $\theta \in \Theta$, $X_\theta^x(\omega, t)$ is cadlag in $t$, and $\theta \rightarrow \hat{X}_\theta^x(\omega, \cdot)$ seen as a mapping $\Theta \rightarrow (D[0, T], \| \cdot \|_\infty)$ is uniformly continuous on compacts. Furthermore, for any $p \geq 2$ there exists $b_p \in (0, \infty)$ s.t.*

$$\sup_{\theta \in \Theta} E \sup_{t \in [0, T]} |X_\theta^x(t)|^p \leq b_p^p (|x| + 1)^p.$$

*Remark.* Theorem K is an extension of Kunita [11, Theorem 3.3.1] using the a.s. version of Kolmogorov's continuity criterion; see Corollary 1 of Protter [19, Theorem 73].

To guarantee that Theorem 1' holds, the requirement that Assumption 5 holds for all $p \geq 2$ can be relaxed to all $p \leq n + \epsilon$ where $n$ is the dimension of $\theta$. However, the intended application of our theory focuses on a regime where $n \gg d$. So, we adopted this stronger version of Assumption 5. This also clarifies the presentations of the following assumptions: To guarantee the main results of this paper, a weaker version of this assumption requires that the assumption holds for all $p \leq 4m + \epsilon$, where $m$ is the growth rate of $v, r, g$, and their derivatives as in Assumption 8 and 7.

Next, we present additional regularities that implies the probabilistic representation in Theorem 1'.

**Assumption 6.** *For each $s \in [0,T]$ and $z \in \mathbb{R}^{d'}$, the mappings $(\theta, x) \rightarrow \mu_\theta(s,x), \sigma_\theta(s,x), \chi_\theta(s,x,z), \rho_\theta(s,x), g_\theta(s,x)$ are $C^{1,0}(\Theta, \mathbb{R}^d)$.*

**Assumption 7.** *The measurable reward rate $\rho_\theta$ and terminal reward $g_\theta$ functions are Lipschitz in $\theta$ in the following sense:*

1. *There exist $m \geq 1$, $\alpha > 1$, and $\left\{ \kappa^\alpha_{\theta,\theta'}(s) \in \mathbb{R}_{>0} : s \in [0,T], \theta, \theta' \in \Theta \right\}$ s.t.*

$$\left( \int_0^T \kappa^\alpha_{\theta,\theta'}(s)ds \right)^{\frac{1}{\alpha}} \leq \ell_\alpha |\theta - \theta'|$$

*for all $\theta, \theta' \in \Theta$, and the reward rate satisfies*

$$|\rho_\theta(s,x) - \rho_{\theta'}(s,x)|^\alpha \leq \kappa^\alpha_{\theta,\theta'}(s)(|x| + 1)^{m\alpha}$$

*for all $s \in [0,T]$, $\theta, \theta' \in \Theta$, $x \in \mathbb{R}^d$.*

2. *For some $\ell \geq 0$, the terminal reward satisfies*

$$|g_\theta(x) - g_{\theta'}(x)| \leq \ell|\theta - \theta'|(|x| + 1)^m.$$

*for all $s \in [0,T]$, $\theta, \theta' \in \Theta$, $x \in \mathbb{R}^d$.*

Note that for notation simplicity, w.l.o.g. we use the same $\kappa^\alpha_{\theta,\theta'}$ and $\ell_\alpha$ as in part 3 of Assumption 5 to denote the Lipschitz coefficient, and the same $m$ as in Assumption 9.

*Remark.* It is not hard to see that Assumptions 6 along with 5 and 7 are generalization of Assumption 1; i.e. if Assumption 1 holds then so will Assumptions 6, 5, and 7.

Next, we slightly generalize the growth part in Assumption 2 as in Assumption 9.

**Assumption 8.** *Assume that $\left\{ v_\theta \in C^{1,2}([0,T], \mathbb{R}^d) : \theta \in \Theta \right\}$ is a family of classical solution to the PIDEs*

$$\partial_t v_\theta + \mathcal{L}_\theta v_\theta + \rho_\theta = 0$$
$$v_\theta(T, \cdot) = g_\theta$$

*where $\mathcal{L}_\theta = \mathcal{L}^C_\theta + \mathcal{L}^J_\theta$ as defined in (3.2).*

**Assumption 9.** *There exists $0 < c_v < \infty$ and $m \geq 1$ s.t.*

$$\sup_{x \in \mathbb{R}^d, t \in [0,T]} \frac{|v_0(t,x)|}{(|x|+1)^m} \leq c_v, \quad \sup_{x \in \mathbb{R}^d, t \in [0,T]} \frac{|\nabla v_0(t,x)|}{(|x|+1)^m} \leq c_v, \quad and \quad \sup_{x \in \mathbb{R}^d, t \in [0,T]} \frac{|H[v_0](t,x)|}{(|x|+1)^m} \leq c_v.$$

*Moreover, for each $\theta$, there exists $c_{\theta,v}$ s.t.*

$$\sup_{x \in \mathbb{R}^d, t \in [0,T]} \frac{|\nabla v_\theta(t,x)|}{(|x|+1)^m} \leq c_{\theta,v}$$

**Theorem 1'.** *If Assumptions 5, 6, 7, 8, and 9 are in force, then the statement in Theorem 1 hold; i.e.*

$$\nabla_\theta v_0(0,x) = E\left[ \int_0^T \nabla_\theta \mathcal{L}_0 v_0(s, X_0^x(s)) + \nabla_\theta \rho_\theta(X_0^x(s))ds + \nabla_\theta g_\theta(X_0^x(T)) \right]$$

*where $\nabla_\theta \mathcal{L}_0 := \nabla_\theta \mathcal{L}^C_0 + \nabla_\theta \mathcal{L}^J_0$ are define in (3.3) and (3.4), respectively.*

## A.2  Probabilistic Representation of the Space Derivatives

Following the same spirit, in this section, we develop a generalized version of Theorem 2, weakening the Assumption 3 to the following Assumption:

**Assumption 10.** *For each $s \in [0,T]$, $z \in \mathbb{R}_0^{d'}$, the coefficients $x \to (\mu_0(s,x), \sigma_0(s,x), \chi_0(s,x,z))$ is second continuously differentiable. Moreover, for each $i,j = 1,\ldots,d$, the coefficients and derivatives, seen as functions $(s,x) \to (\alpha(s,x), \beta(s,x), \zeta(s,x,\cdot))$ where $(\alpha, \beta, \zeta) = (\mu_0, \sigma_0, \chi_0)$, $(\partial_i \mu_0, \partial_i \sigma_0, \partial_i \chi_0)$, and $(\partial_j \partial_i \mu_0, \partial_j \partial_i \sigma_0, \partial_j \partial_i \chi_0)$ satisfies the following conditions:*

1. *At $x = 0$, the coefficients are uniformly bounded in time: for all $p \geq 2$,*

$$\sup_{s \in [0,T]} \left[ |\alpha(s,0)| + |\beta(s,0)| + \int_{\mathbb{R}_0^{d'}} \left| \frac{\zeta(s,0,z)}{\gamma(z)} \right|^p \mu(dz) \right] < \infty$$

2. *The coefficients are uniform Lipschitz in $x$: There exists constants $c$ and $\{c_p : p \geq 2\}$ s.t.*
$$|\alpha(s,x) - \alpha(s,x')| \leq c\,|x - x'|, \quad |\beta(s,x) - \beta(s,x')| \leq c\,|x - x'|,$$
   *and for any $p \geq 2$*

$$\left( \int_{\mathbb{R}_0^{d'}} \left| \frac{\zeta(s,x,z)}{\gamma(z)} - \frac{\zeta(s,x',z)}{\gamma(z)} \right|^p \mu(dz) \right)^{\frac{1}{p}} \leq c_p |x - x'|$$

   *for all $s \in [0,T]$, $x, x' \in \mathbb{R}^d$.*

**Proposition A.1.** *Suppose that Assumption 10 is in force. Then, the family of stochastic flow solutions $\left\{ X_0^x(s,t), 0 \leq s \leq t \leq T : x \in \mathbb{R}^d \right\}$ of the SDEs (3.1) has a version $\hat{X}_0$ that is second differentiable in $x$ at any time. Moreover, $\left\{ \hat{X}_0^x, \nabla \hat{X}_0^x, H[\hat{X}_0^x] : x \in \mathbb{R}^d \right\}$ is a version of the solutions of the systems of SDEs (3.1) and (3.5). Further, the family of solutions of (3.1) and (3.5) satisfies the following properties:*

1. *For each $p \geq 1$, there is $0 < b_p < \infty$ s.t.*
$$E \sup_{t \in [0,T]} |X_0^x(t)|^p \leq b_p^p (|x| + 1)^p$$

   *and the derivatives*
$$\sup_{x \in \mathbb{R}^d} E \sup_{0 \leq s \leq t \leq T} |\nabla X_0^x(s,t)|^p \leq b_p^p, \qquad \sup_{x \in \mathbb{R}^d} E \sup_{0 \leq s \leq t \leq T} |H[X_0^x](s,t)|^p \leq b_p^p.$$

2. *For any $p \geq 1$, there exists $0 < l_p < \infty$ s.t. for all $x, x' \in \mathbb{R}^d$,*
$$E \sup_{0 \leq s \leq t \leq T} |X_0^x(s,t) - X_0^{x'}(s,t)|^p \leq l_p^p |x - x'|^p,$$
$$E \sup_{0 \leq s \leq t \leq T} |\nabla X_0^x(s,t) - \nabla X_0^{x'}(s,t)|^p \leq l_p^p |x - x'|^p.$$

A proof of Proposition A.1 is provided in Appendix C.

*Remark.* We observe that part 2 of Assumption 10 will imply the space derivatives of the coefficients are bounded, which is used to get the $L^p$ boundedness and Lipschitzness of the derivative processes. Assumption 10 also ensures that the second derivative is uniformly Lipschitz as well. This is not used in the proof for the upcoming results.

We establish the following Theorem 2' generalizing 2. The proof is deferred to Appendix D.

**Theorem 2'.** *Under Assumptions 10 and 4, then (2.6) holds; i.e.*

$$\nabla v_0(t,x)^\top = E \left[ \int_t^T \nabla \rho_0^\top \nabla X_0^x(t,r) dr + \nabla g_0^\top \nabla X_0^x(t,T) \right],$$

$$H[v_0](t,x) = E \left[ \int_t^T \nabla X_0^x(t,r)^\top H[\rho_0] \nabla X_0^x(t,r) + \left\langle \nabla \rho_0, H[X_{0,\cdot}^x](t,r) \right\rangle dr \right]$$
$$+ E \left[ \nabla X_0^x(t,T)^\top H[g_0] \nabla X_0^x(t,T) + \left\langle \nabla g_0, H[X_{0,\cdot}^x](t,T) \right\rangle \right].$$

*where $\{X_0^x, \nabla X_0^x, H[X_0^x]\}$ are the strong solutions to (3.1) and (3.5).*

## A.3 The Estimator

With Proposition A.1 and Theorem 2', we are ready to define our generator gradient estimator for $\nabla_\theta v_0(0, x)$ and show that it is unbiased and has a variance that grows polynomially in $x$.

First, recall the definition of $Z$ and $H$ in (2.6)

$$Z(t,x)^\top := \int_t^T \nabla \rho_0^\top \nabla X_0^x(t,r)dr + \nabla g_0^\top \nabla X_0^x(t,T),$$

$$H(t,x) := \int_t^T \nabla X_0^x(t,r)^\top H[\rho_0]\nabla X_0^x(t,r) + \left\langle \nabla \rho_0, H[X_{0,.}^x](t,r)\right\rangle dr$$

$$+ \nabla X_0^x(t,T)^\top H[g_0]\nabla X_0^x(t,T) + \left\langle \nabla g_0, H[X_{0,.}^x](t,T)\right\rangle.$$

Observe that by Theorem 2', the integrability in Proposition A.1,

$$E \int_0^T \partial_{\theta_k}\mu_0(t, X_0^x(0,t))^\top Z(t, X_0^x(0,t))dt$$

$$= \int_0^T dt E \left[ \partial_{\theta_k}\mu_0(t, X_0^x(0,t))^\top E\left[\int_t^T \nabla \rho_0^\top \nabla X_0^{X_0^x(0,t)}(t,r)dr + \nabla g_0^\top \nabla X_0^{X_0^x(0,t)}(t,T)\bigg| X_0^x(0,t)\right]\right]$$

$$= E \int_0^T \partial_{\theta_k}\mu_0(t, X_0^x(0,t))^\top \nabla v_0(t, X_0^x(0,t))dt;$$

(A.1)

a similar property hold for the $H(t,x)$ process as well. Therefore, we can replace the derivatives $\nabla v_0$ with $Z$ and $H[v_0]$ with $H$ in Theorem 1' without changing the expectation, showing the validity of (2.8). In particular, this implies that the generator gradient estimator defined in (2.9) is unbiased.

Finally, we establish a generalized version Theorem 3 using the assumptions in this section. The additional proof of this theorem is presented in Appendix E.

**Theorem 3'.** *Suppose Assumptions 4-10 are in force. Then, the generator gradient estimator* (2.9) *is unbiased; i.e.* $ED(x) = \nabla_\theta v_0(0,x)$. *Moreover, if the* $\alpha > 1$ *in item 1 of Assumption 7 is replaced by* $\alpha > 2$, *then the variance* $\mathrm{Var}(D(x)) \le C(|x|+1)^{2m+4}$ *has at most polynomial growth in* $x$, *where* $C$ *can be dependent on other parameters of the problem but not* $x$.

*Remark.* The $m$ signifies the growth rate of the rewards and their derivatives. The extra additive factor 2 in the variance is from the growth of the $\theta$ derivative of $a_0$, the volatility squared.

## B  Proof of Theorem 1'

In this section, we prove Theorem 1' and hence the simplified Theorem 1.

*Proof of Theorem 1'.* First, we recall Itô's formula. For $f \in C^{1,2}([0,T] \times \mathbb{R}^d)$,

$$df(t, X_\theta^x(t)) = \left[\partial_t f(t, X_\theta^x(t)) + (\mathcal{L}_\theta^C f)(t, X_\theta^x(t))\right]dt + \sum_{i=1}^d \sum_{k=1}^{d'} \partial_i f(t, X_\theta^x(t-))\sigma_{\theta,i,k}(t, X_\theta^x(t))dB_k(t)$$

$$+ (\mathcal{L}_\theta^J f)(t, X_\theta^x(t))dt + \int_{\mathbb{R}_0^{d'}} \left[f(t, X_\theta^x(t-) + \chi_\theta(t, X_\theta^x(t-),z)) - f(t, X_\theta^x(t-))\right]d\widetilde{N}(dt, dz),$$

(B.1)

where the operators $\mathcal{L}_\theta^C$ and $\mathcal{L}_\theta^J$ are defined in (3.2).

Then, an application of Itô's formula (B.1) under Assumption 8 and 9 yields the following result for which the proof is presented in Section B.1.

**Lemma 1.** *For any* $\theta \in \Theta$,

$$M_{\theta,\theta}(t) = v_\theta(t, X_\theta^x(t)) - v_\theta(0,x) + \int_0^t \rho_\theta(s, X_\theta^x(s))ds$$

$$M_{0,\theta}(t) = v_0(t, X_\theta^x(t)) - v_0(0,x) - \int_0^t \partial_s v_0(s, X_\theta^x(s)) + \mathcal{L}_\theta v_0(s, X_\theta^x(s))ds$$

*are martingales for* $0 \le t \le T$.

Therefore,

$$0 = E[M_{\theta,\theta}(T) - M_{0,\theta}(T)].$$

Then, rearranging terms, one gets

$$v_\theta(0,x) - v_0(0,x)$$

$$= Ev_\theta(T, X_\theta^x(T)) - v_0(T, X_\theta^x(T)) + \int_0^T \rho_\theta(s, X_\theta^x(s)) \pm \rho_0(s, X_\theta^x(s)) + \partial_s v_0(s, X_\theta^x(s)) + \mathcal{L}_\theta v_0(s, X_\theta^x(s))ds$$

$$\stackrel{(i)}{=} Eg_\theta(X_\theta^x(T)) - g_0(X_\theta^x(T)) + \int_0^T \rho_\theta(s, X_\theta^x(s)) - \rho_0(s, X_\theta^x(s)) + \mathcal{L}_\theta v_0(s, X_\theta^x(s)) - \mathcal{L}_0 v_0(s, X_\theta^x(s))ds$$

$$\stackrel{(ii)}{=} E\int_0^T \left(\mathcal{L}_\theta^C - \mathcal{L}_0^C\right) v_0(s, X_\theta^x(s))ds + \int_0^T \left(\mathcal{L}_\theta^J - \mathcal{L}_0^J\right) v_0(s, X_\theta^x(s))ds$$

$$+ Eg_\theta(X_\theta^x(T)) - g_0(X_\theta^x(T)) + \int_0^T \rho_\theta(s, X_\theta^x(s)) - \rho_0(s, X_\theta^x(s))ds$$

(B.2)

where $(i)$ follows from Assumption 8 that $v_\theta(T, \cdot) = g_\theta(\cdot)$ and $\rho_0(s,x) = -\partial_s v_0(s,x) - \mathcal{L}_0 v_0(s,x)$ for all $x \in \mathbb{R}^d$ and $0 \leq s < T$, and $(ii)$ recalls the definition that $\mathcal{L}_\theta = \mathcal{L}_\theta^C + \mathcal{L}_\theta^J$.

To conclude Theorem 1', we analyze the finite difference approximations of the above three expectations, where they correspond to the **continuous**, the **jump**, and the **rewards** part, respectively. The results are summarized by the following Proposition B.1, whose proof is deferred to Appendix B.2.

**Proposition B.1.** *Under the assumptions of Theorem 1', for $K = C, J$,*

$$\lim_{\theta \to 0} \frac{1}{|\theta|} \left| E\left[\int_0^T \left(\mathcal{L}_\theta^K - \mathcal{L}_0^K\right) v_0(s, X_\theta^x(s))ds\right] - \theta^T E\left[\int_0^T \nabla_\theta \mathcal{L}_0^K v_0(s, X_0^x(s))ds\right]\right| = 0.$$

*where $\nabla_\theta \mathcal{L}_0^C$ and $\nabla_\theta \mathcal{L}_0^J$ are defined in (3.3) and (3.4) respectively. Moreover,*

$$\lim_{\theta \to 0} \frac{1}{|\theta|} \left| E\left[\int_0^T \rho_\theta(s, X_\theta^x(s)) - \rho_0(s, X_\theta^x(s))ds\right] - \theta^T E\left[\int_0^T \nabla_\theta \rho_0(s, X_0^x(s))ds\right]\right| = 0$$

*and*

$$\lim_{\theta \to 0} \frac{1}{|\theta|} \left| E\left[g_\theta(X_\theta^x(T)) - g_0(X_\theta^x(T))\right] - \theta^T E\nabla_\theta g_0(X_0^x(T))\right| = 0.$$

With Proposition B.1 handling each term in (B.2), we conclude that

$$\lim_{\theta \to 0} \frac{1}{|\theta|} \left| v_\theta(0,x) - v_0(0,x) - \theta^T E\left[\int_0^T \nabla_\theta \mathcal{L}_0 v_0(s, X_0^x(s)) + \nabla_\theta \rho_0(X_0^x(s))ds + \nabla_\theta g_0(X_0^x(T))\right]\right| = 0$$

$\square$

## B.1   Proof of Lemma 1

Apply Itô's formula (B.1) to $v_\theta(t, X_\theta^x(t))$ yield

$$0 = v_\theta(t, X_\theta^x(t)) - v_\theta(0,x) - \int_0^t \partial_s v_\theta(t, X_\theta^x(s)) - \mathcal{L}_\theta v_\theta(t, X_\theta^x(s))dt$$

$$- \sum_{i=1}^d \sum_{k=1}^{d'} \int_0^t \partial_i v_\theta(s, X_\theta^x(s-))\sigma_{\theta,i,k}(s, X_\theta^x(s-))dB_k(t)$$

$$- \int_0^t \int_{\mathbb{R}_0^{d'}} [v_\theta(s, X_\theta^x(s-) + \chi_\theta(s, X_\theta^x(s-), z)) - v_\theta(s, X_\theta^x(s-))] d\widetilde{N}(ds, dz)$$

$$=: M_{\theta,\theta}(t) - I_1(t) - I_2(t)$$

by Assumption 8, where $I_1(t)$ and $I_2(t)$ denotes the Itô stochastic integrals on the previous lines, respectively. Since the integrands are a.s. finite, $I_1(t)$ and $I_2(t)$ are local martingales. We show that they are true martingales. First, for $I_1$, apply the Burkholder-Davis-Gundy inequality

$$E|I_1(t)|^2 \leq E \sup_{t \leq T} |I_1(t)|^2$$

$$\leq C \sum_{i=1}^{d} \sum_{k=1}^{d'} E \int_0^T |\partial_i v_\theta(s, X_\theta^x(s)) \sigma_{\theta,i,k}(s, X_\theta^x(s))|^2 \, ds$$

$$\leq CE \int_0^T |\nabla v_\theta(s, X_\theta^x(s))|^2 \, |\sigma_\theta(s, X_\theta^x(s))|^2 \, ds$$

$$\leq CE \int_0^T |\nabla v_\theta(s, X_\theta^x(s))|^4 \, ds E \int_0^T |\sigma_\theta(s, X_\theta^x(s)) - \sigma_\theta(s, 0) + \sigma_\theta(s, 0)|^4 \, ds$$

where $C$ is some constant that could change line by line. Notice that by Assumption 5,

$$\sup_{\theta \in \Theta, t \in [0,T]} |\sigma_\theta(t, 0)| =: \sigma_\vee < \infty. \tag{B.3}$$

Therefore, by Assumption 5 item 2, Assumption 9, and Theorem K

$$E|I_1(t)|^2 \leq CE \int_0^T (|X_\theta^x(s)| + 1)^{4m} ds E \int_0^T (|X_\theta^x(s)| + \sigma_\vee)^4 ds < \infty.$$

Therefore, $I_1$ is a martingale. For $I_2$, by Kunita [11, Proposition 2.6.1]

$$E|I_2(t)|^2 \leq CE \int_0^t \int_{\mathbb{R}_0^{d'}} \left( \frac{v_\theta(s, X_\theta^x(s) + \chi_\theta(s, X_\theta^x(s), z)) - v_\theta(s, X_\theta^x(s-))}{\gamma(z)} \right)^2 \mu(dz) ds$$

$$\overset{(i)}{=} CE \int_0^t \int_{\mathbb{R}_0^{d'}} \left( \frac{\nabla v_\theta(s, X_\theta^x(s) + \xi \chi_\theta(s, X_\theta^x(s), z))^\top \chi_\theta(s, X_\theta^x(s), z)}{\gamma(z)} \right)^2 \mu(dz) ds$$

$$\overset{(ii)}{=} CE \int_0^t \int_{\mathbb{R}_0^{d'}} \left| \frac{\chi_\theta(s, X_\theta^x(s), z)}{\gamma(z)} \right|^4 \mu(dz) ds \cdot E \int_0^t \int_{\mathbb{R}_0^{d'}} |\nabla v_\theta(s, X_\theta^x(s) + \xi \chi_\theta(s, X_\theta^x(s), z))|^4 \mu(dz) ds$$

where $(i)$ follows from the mean value theorem with $\xi := \xi_\theta(s, X_\theta^x(s), z) \in [0, 1]$, and $(ii)$ follows from the Cauchy-Schwartz inequality applied to the integral w.r.t. the finite measure $P \times \text{Leb} \times \mu$. By Assumption 5,

$$\chi_{\theta, \vee}^4 := \sup_{s \in [0,T]} \int_{\mathbb{R}_0^{d'}} \frac{|\chi_\theta(s, 0, z)|^4}{\gamma(z)^4} \mu(dz) < \infty$$

Again, by Assumption 5 item 2 and Theorem K,

$$E \int_0^t \int_{\mathbb{R}_0^{d'}} \left| \frac{\chi_\theta(s, X_\theta^x(s), z)}{\gamma(z)} \right|^4 \mu(dz) ds \leq CE \int_0^t \int_{\mathbb{R}_0^{d'}} \left| \frac{\chi_\theta(s, X_\theta^x(s), z) - \chi_\theta(s, 0, z)}{\gamma(z)} \right|^4 \mu(dz) ds + C \chi_{\theta, \vee}^4$$

$$\leq C \left( \chi_{\theta, \vee}^4 + E \int_0^t |X_\theta^x(s)|^4 ds \right)$$

$$< \infty.$$

Also, by Assumption 9 and $\xi \in [0, 1]$

$$E \int_0^t \int_{\mathbb{R}_0^{d'}} |\nabla v_\theta(s, X_\theta^x(s) + \xi \chi_\theta(s, X_\theta^x(s), z))|^4 \mu(dz) ds$$

$$\leq c_{\theta, v} E \int_0^t \int_{\mathbb{R}_0^{d'}} (|X_\theta^x(s)| + |\chi_\theta(s, X_\theta^x(s), z)| + 1)^{4m} \mu(dz) ds$$

$$\overset{(i)}{\leq} C \left( E \int_0^t |X_\theta^x(s)|^{4m} ds + E \int_0^t \int_{\mathbb{R}_0^{d'}} \frac{|\chi_\theta(s, X_\theta^x(s), z))|^{4m}}{\gamma(z)^{4m}} \mu(dz) ds + 1 \right)$$

$$< \infty$$

where $(i)$ follows from $\mu$ being a finite measure and $\gamma(z) = |z| \wedge 1 \leq 1$. This shows that $I_2$, hence $M_{\theta,\theta}(t)$ is a martingale.

To show that $M_{0,\theta}(t)$ is a martingale, we employ the same derivation with $v_\theta$ replaced by $v_0$. This completes the proof of Lemma 1.

## B.2 Proof of Proposition B.1

Since $X^x$ is indistinguishable from $\hat{X}^x$, we can use $X^x$ and $\hat{X}^x$ interchangeably when evaluating expectations. Therefore, it is understood that we use $\hat{X}^x$ when we need continuity in $\theta$, while we keep the notation $X^x$.

In this proof, for notation simplicity, the letter $C$ will denote a constant that could change from line to line. $C$ can be dependent on the dimensions $d, d'$, the growth rate $m$, the horizon $T$, the Lévy measure $\nu$, and polynomial power $p$ or $\alpha$. But it doesn't depend on $\theta$ (or sometimes $\delta$) and $x$.

**The Continuous Part:** We prove the claim that the derivative for the continuous part should be

$$E \int_0^T \nabla_\theta \mathcal{L}_0^C v_0(s, X_0^x(s)) ds, \tag{B.4}$$

where $\nabla_\theta \mathcal{L}_0$ is defined in (3.3).

To proceed, we also claim that

$$E \left| \int_0^T \left( \mathcal{L}_\theta^C - \mathcal{L}_0^C \right) v_0(s, X_\theta^x(s)) ds \right| < \infty, \quad E \left| \int_0^T \nabla_\theta \mathcal{L}_0^C v_0(s, X_0^x(s)) ds \right| < \infty \tag{B.5}$$

so that the derivative ratio and the derivative are well-defined. The finiteness of these expectations is shown below.

To prove the claimed expression (B.4) is indeed the derivative, we consider the limit

$$\lim_{\theta \to 0} \frac{1}{|\theta|} \left| E \left[ \int_0^T \left( \mathcal{L}_\theta^C - \mathcal{L}_0^C \right) v_0(s, X_\theta^x(s)) ds \right] - \theta^T E \left[ \int_0^T \nabla_\theta \mathcal{L}_0^C v_0(s, X_0^x(s)) ds \right] \right|$$
$$\leq T \lim_{\theta \to 0} E \frac{1}{T} \int_0^T \frac{1}{|\theta|} \left| \left( \mathcal{L}_\theta^C - \mathcal{L}_0^C \right) v_0(s, X_\theta^x(s)) - \theta^T \nabla_\theta \mathcal{L}_0^C v_0(s, X_0^x(s)) \right| ds \tag{B.6}$$

To show the r.h.s. go to 0, we first show what's inside the two integrals is U.I. Consider for $\alpha > 1$,

$$E \frac{1}{T} \int_0^T \frac{1}{|\theta|^\alpha} \left| \left( \mathcal{L}_\theta^C - \mathcal{L}_0^C \right) v_0(s, X_\theta^x(s)) - \theta^T \nabla_\theta \mathcal{L}_0^C v_0(s, X_0^x(s)) \right|^\alpha ds$$
$$\leq 2^{\alpha-1} E \frac{1}{T} \int_0^T \frac{1}{|\theta|^\alpha} \left| \left( \mathcal{L}_\theta^C - \mathcal{L}_0^C \right) v_0(s, X_\theta^x(s)) \right|^\alpha + 2^{\alpha-1} E \frac{1}{T} \int_0^T \left| \nabla_\theta \mathcal{L}_0^C v_0(s, X_0^x(s)) \right|^\alpha ds \tag{B.7}$$

For the first term, consider

$$\left| \left( \mathcal{L}_\theta^C - \mathcal{L}_0^C \right) v_0(t, x) \right|$$
$$= (\mu_\theta(t, x) - \mu_0(t, x))^\top \nabla_x v_0(t, x) + \sum_{i,j=1}^d (a_{\theta,i,j}(t, x) - a_{0,i,j}(t, x)) \partial_i \partial_j v_0(t, x).$$
$$\leq |\mu_\theta(t, x) - \mu_0(t, x)| |\nabla_x v_0(t, x)| + |a_\theta(t, x) - a_0(t, x)| |H[v_0](t, x)| \tag{B.8}$$
$$\overset{(i)}{\leq} c_v (|x| + 1)^m (|\mu_\theta(t, x) - \mu_0(t, x)| + |a_\theta(t, x) - a_0(t, x)|)$$

where $(i)$ follows from Assumption 9. For the second term, recall the definition in (2.1):

$$|a_\theta(t,x) - a_0(t,x)|$$

$$\leq \sum_{i,j=1}^{d} |a_{\theta,i,j}(t,x) - a_{0,i,j}(t,x)|$$

$$= \frac{1}{2} \sum_{i,j=1}^{d} \left| \sum_{k=1}^{d'} \sigma_{\theta,i,k}(t,x)\sigma_{\theta,j,k}(t,x) - \sigma_{0,i,k}(t,x)\sigma_{0,j,k}(t,x) \right|$$

$$= \frac{1}{2} \sum_{i,j=1}^{d} \left| \sum_{k=1}^{d'} \left[\sigma_{\theta,i,k}(t,x) - \sigma_{0,i,k}(t,x)\right] \sigma_{\theta,j,k}(t,x) + \sigma_{0,i,k}(t,x) \left[\sigma_{\theta,j,k}(t,x) - \sigma_{0,j,k}(t,x)\right] \right|$$

$$\leq \frac{1}{2} \sum_{i,j=1}^{d} \sum_{k=1}^{d'} |\sigma_{\theta,i,k}(t,x) - \sigma_{0,i,k}(t,x)| \, |\sigma_{\theta,j,k}(t,x)| + |\sigma_{0,i,k}(t,x)| \, |\sigma_{\theta,j,k}(t,x) - \sigma_{0,j,k}(t,x)|$$

The two terms can be handled in the same way as follows:

$$\frac{1}{2} \sum_{i,j=1}^{d} \sum_{k=1}^{d'} |\sigma_{\theta,i,k}(t,x) - \sigma_{0,i,k}(t,x)| \, |\sigma_{\theta,j,k}(t,x)|$$

$$\leq \frac{1}{2} \sum_{j=1}^{d} |\sigma_{\theta,j,\cdot}(t,x)| \sum_{i=1}^{d} |\sigma_{\theta,i,\cdot}(t,x) - \sigma_{0,i,k}(t,x)|$$

$$\leq \frac{1}{2} \left( d \sum_{j=1}^{d} \frac{1}{d} \sqrt{\sum_{k=1}^{d'} |\sigma_{\theta,j,\cdot}(t,x)|^2} \right) \left( d \sum_{i=1}^{d} \frac{1}{d} \sqrt{\sum_{k=1}^{d'} |\sigma_{\theta,i,k}(t,x) - \sigma_{0,i,k}(t,x)|^2} \right)$$

$$\leq \frac{d}{2} \sqrt{\sum_{j=1}^{d} \sum_{k=1}^{d'} |\sigma_{\theta,j,k}(t,x)|^2} \sqrt{\sum_{i=1}^{d} \sum_{k=1}^{d'} |\sigma_{\theta,i,k}(t,x) - \sigma_{0,i,k}(t,x)|^2}$$

$$= \frac{d}{2} |\sigma_\theta(t,x)| \, |\sigma_\theta(t,x) - \sigma_0(t,x)|$$

$$\leq \frac{d}{2} \left( |\sigma_\theta(t,x) - \sigma_\theta(t,0)| + |\sigma_\theta(t,0)| \right) |\sigma_\theta(t,x) - \sigma_0(t,x)|$$

$$\overset{(i)}{\leq} \frac{d}{2} \left( c|x| + \sigma_\vee \right) |\sigma_\theta(t,x) - \sigma_0(t,x)|.$$

where $(i)$ follows from Assumption 5 item 2 and the constant bound for $\sigma_\theta(t,0)$ in (B.3). Going back to inequality (B.7), these bounds implies that

$$\left| \left(\mathcal{L}_\theta^C - \mathcal{L}_0^C\right) v_0(t,x) \right|^\alpha \leq 2^{\alpha-1} c_v (|x|+1)^{m\alpha} \left( |\mu_\theta(t,x) - \mu_0(t,x)|^\alpha + |a_\theta(t,x) - a_0(t,x)|^\alpha \right)$$

$$\leq C(|x|+1)^{m\alpha} \left( |\mu_\theta(t,x) - \mu_0(t,x)|^\alpha + (c|x| + \sigma_\vee)^\alpha |\sigma_\theta(t,x) - \sigma_0(t,x)|^\alpha \right)$$

$$\overset{(i)}{\leq} C(|x|+1)^{m\alpha} \left( \kappa_{\theta,0}^\alpha(t)(|x|+1)^\alpha + \kappa_{\theta,0}^\alpha(t)(|x|+1)^{2\alpha} \right)$$

$$\leq C\kappa_{\theta,0}^\alpha(t)(|x|+1)^{(m+2)\alpha}$$

for some $C$ that doesn't depend on $\theta$, where $(i)$ follows from item 3 of Assumption 5. Therefore,

$$
\frac{1}{|\theta|^\alpha} E \frac{1}{T} \int_0^T \left| \left( \mathcal{L}_\theta^C - \mathcal{L}_0^C \right) v_0(s, X_\theta^x(s)) \right|^\alpha ds
$$

$$
\leq \frac{C}{|\theta|^\alpha} E \frac{1}{T} \int_0^T \kappa_{\theta,0}^\alpha(s)(|X_\theta^x(s)| + 1)^{(m+2)\alpha} ds
$$

$$
\overset{(i)}{=} \frac{C}{|\theta|^\alpha} \frac{1}{T} \int_0^T \kappa_{\theta,0}^\alpha(s) E(|X_\theta^x(s)| + 1)^{(m+2)\alpha} ds
$$

$$
\leq \frac{C}{|\theta|^\alpha} \int_0^T \kappa_{\theta,0}^\alpha(s) ds \sup_{\theta \in \Theta, s \in [0,T]} E(|X_\theta^x(s)| + 1)^{(m+2)\alpha} \tag{B.9}
$$

$$
\leq C l_\alpha^\alpha \sup_{\theta \in \Theta} 2^{(m+2)\alpha-1} \left( E \sup_{s \in [0,T]} |X_\theta^x(s)|^{(m+2)\alpha} + 1 \right)
$$

$$
\overset{(ii)}{\leq} C \left( b_{(m+2)\alpha}^{(m+2)\alpha} (|x| + 1)^{(m+2)\alpha} + 1 \right)
$$

$$
\leq C(|x| + 1)^{(m+2)\alpha}
$$

where $(i)$ applies Fubini's theorem due to the positivity of $\kappa_{\theta,0}^\alpha$, and $(ii)$ follows from Theorem K. We have shown that this expectation above is finite and independent of $\theta$. Note that, in particular, this implies that the first expectation in (B.5) is finite as well.

For the second term in the last line of (B.7), we first consider for matrix $M \in \mathbb{R}^{n \times d}$ and vector $v \in \mathbb{R}^d$,

$$
|Mv|^\alpha = \left( \sum_{l=1}^n \left| \sum_{i=1}^d M_{l,i} v_i \right|^2 \right)^{\alpha/2}
$$

$$
\leq \|v\|_\infty^\alpha \left( \sum_{l=1}^n \left( \sum_{i=1}^d |M_{l,i}| \right)^2 \right)^{\alpha/2}
$$

$$
\leq |v|^\alpha \left( \sum_{l=1}^n \sum_{i=1}^d |M_{l,i}| \right)^\alpha
$$

$$
\leq (nd)^{\alpha-1} |v|^\alpha \sum_{l=1}^n \sum_{i=1}^d |M_{l,i}|^\alpha
$$

Apply this inequality, we obtain

$$
E \frac{1}{T} \int_0^T \left| \nabla_\theta \mathcal{L}_0^C v_0(s, X_0^x(s)) \right|^\alpha ds
$$

$$
\leq 2^{\alpha-1} E \frac{1}{T} \int_0^T \left| \sum_{i=1}^d \nabla_\theta \mu_{0,i}(s, X_0^x(s)) \partial_i v_0(s, X_0^x(s)) \right|^\alpha + \left| \sum_{i,j=1}^d \nabla_\theta a_{0,i,j}(s, X_0^x(s)) \partial_i \partial_j v_0(s, X_0^x(s)) \right|^\alpha ds
$$

$$
\overset{(i)}{\leq} C E \frac{1}{T} \int_0^T (|X_0^x(s)| + 1)^{(m+1)\alpha} \left( \sum_{l=1}^n \sum_{i=1}^d |\partial_{\theta_l} \mu_{0,i}(s, X_0^x(s))|^\alpha + \sum_{l=1}^n \sum_{i,j=1}^d |\partial_{\theta_l} a_{0,i,j}(s, X_0^x(s))|^\alpha \right) ds
$$

$$
\overset{(ii)}{\leq} C \left[ E \frac{1}{T} \int_0^T \sum_{l=1}^n \sum_{i=1}^d |\partial_{\theta_l} \mu_{0,i}(s, X_0^x(s))|^{2\alpha} + \sum_{l=1}^n \sum_{i,j=1}^d |\partial_{\theta_l} a_{0,i,j}(s, X_0^x(s))|^{2\alpha} ds \right]^{1/2}
$$

$$
\tag{B.10}
$$

where $(i)$ uses Assumption 9 and the previous matrix norm inequality, and $(ii)$ uses Cauchy-Schwartz inequality. Let $e_l \in \mathbb{R}^n$ be the unit vector with the $l$'th coordinate equal to 1. Now by Assumption 5,

we have that for fixed $\epsilon > 0$ and $l = 1, 2, \ldots, n$

$$E\frac{1}{T}\int_0^T \frac{|\mu_{\delta e_l,i}(s, X_0^x(s)) - \mu_{0,i}(s, X_0^x(s))|^{2\alpha+\epsilon}}{\delta^{2\alpha+\epsilon}}ds \leq \delta^{-2\alpha-\epsilon}E\frac{1}{T}\int_0^T \kappa_{\delta e_l,0}^{2\alpha+\epsilon}(s)(1+|X_0^x(s)|)^{2\alpha+\epsilon}ds.$$

By the same argument as in (B.9), this is uniformly bounded in $\delta$. Hence

$$\begin{aligned}
E\frac{1}{T}\int_0^T |\partial_{\theta_l}\mu_{0,i}(s, X_0^x(s))|^{2\alpha}ds &= E\frac{1}{T}\int_0^T \lim_{\delta\downarrow 0}\left|\frac{\mu_{\delta e_l,i}(s, X_0^x(s)) - \mu_{0,i}(s, X_0^x(s))}{\delta}\right|^{2\alpha}ds \\
&= \lim_{\delta\downarrow 0}E\frac{1}{T}\int_0^T \left|\frac{\mu_{\delta e_l,i}(s, X_0^x(s)) - \mu_{0,i}(s, X_0^x(s))}{\delta}\right|^{2\alpha}ds \\
&\leq \sup_{\theta\in\Theta}\frac{1}{|\theta|^{2\alpha}}E\frac{1}{T}\int_0^T \kappa_{\theta,0}^{2\alpha}(s)(1 + |X_0^x(s)|)^{2\alpha}ds \\
&\leq \psi_1
\end{aligned}$$

(B.11)

where, again, by the same argument as in (B.9), $\psi_1$ is chosen to be finite. For the second term in the last line of (B.10), we consider the quantity

$$\begin{aligned}
\phi(p, \delta, t, x) &:= \sum_{i,j=1}^d \left|\frac{1}{2}\sum_{k=1}^{d'} \sigma_{0,j,k}(t,x)\frac{\sigma_{\delta e_l,i,k}(t,x) - \sigma_{0,i,k}(t,x)}{\delta} + \sigma_{0,i,k}(t,x)\frac{\sigma_{\delta e_l,j,k}(t,x) - \sigma_{0,j,k}(t,x)}{\delta}\right|^p \\
&\overset{(i)}{\leq} \frac{d'^{p-1}}{2}\sum_{i,j=1}^d \sum_{k=1}^{d'} |\sigma_{0,j,k}(t,x) + \sigma_{0,i,k}(t,x)|^p \frac{\kappa_{\delta e_l,0}^p(t)}{\delta^p}(|x| + 1)^p \\
&\overset{(ii)}{\leq} dd'^{p-1}\sqrt{dd'}\frac{\kappa_{\delta e_l,0}^p(t)}{\delta^p}(|x| + 1)^p|\sigma_0(t,x)|^p \\
&\overset{(iii)}{\leq} C\frac{\kappa_{\delta e_l,0}^p(t)}{\delta^p}(|x| + 1)^p(|\sigma_0(t,x) - \sigma_0(t,0)| + \sigma_\vee)^p \\
&\leq C\frac{\kappa_{\delta e_l,0}^p(t)}{\delta^p}(|x| + 1)^{2p}
\end{aligned}$$

(B.12)

where $(i)$ follows from Assumption 5 and $(ii)$ applies Jensen's inequality and $(iii)$ recalls the definition in (B.3). Note that the reason we define $\phi(p, \delta, t, x)$ is because

$$\begin{aligned}
\sum_{i,j=1}^d |\partial_{\theta_l}a_{0,i,j}(t,x)|^{2\alpha} &= \sum_{i,j=1}^d \left|\frac{1}{2}\sum_{k=1}^{d'} \sigma_{0,j,k}(t,x)\partial_{\theta_l}\sigma_{0,i,k}(t,x) + \sigma_{0,i,k}(t,x)\partial_{\theta_l}\sigma_{0,j,k}(t,x)\right|^{2\alpha} \\
&= \lim_{\delta\downarrow 0}\phi(2\alpha, \delta, t, x).
\end{aligned}$$

From (B.12), we see that the same argument in (B.9) implies the $[0,T]\times\Omega$ integrability of $\phi(2\alpha + \epsilon, \delta, s, X_0^x(s))$, uniformly in $\delta$. This then implies that $\phi(2\alpha, \cdot, s, X_0^x(s))$ is U.I. for $\delta$ in a neighbourhood of $0$. Therefore, we see that

$$\begin{aligned}
E\frac{1}{T}\int_0^T \sum_{i,j=1}^d |\partial_{\theta_l}a_{0,i,j}(s, X_\theta^x(s))|^{2\alpha}\,ds &= \lim_{\delta\downarrow 0}E\frac{1}{T}\int_0^T \phi(2\alpha, \delta, s, X_0^x(s))ds \\
&\leq C\sup_{\theta\in\Theta}E\frac{1}{T}\int_0^T \frac{\kappa_{\theta,0}(s)^{2\alpha}}{|\theta|^{2\alpha}}(|X_0^x(s)| + 1)^{4\alpha}ds \\
&\leq \psi_2 < \infty
\end{aligned}$$

(B.13)

Combining these with (B.10), we have establish that

$$E\frac{1}{T}\int_0^T \left|\nabla_\theta\mathcal{L}_0^C v_0(s, X_0^x(s))\right|^\alpha ds \leq C\sqrt{nd\psi_1 + n\psi_2} < \infty.$$

(B.14)

In particular, this shows the second expectation in (B.5) is finite as well.

Therefore, in view of (B.7), (B.9), and (B.14), we conclude that

$$\frac{1}{|\theta|} \left| \left( \mathcal{L}_\theta^C - \mathcal{L}_0^C \right) v_0(s, X_\theta^x(s)) - \theta^T \nabla_\theta \mathcal{L}_0^C v_0(s, X_0^x(s)) \right|$$

is U.I. on $(\Omega \times [0,T], \mathcal{F} \times \mathcal{B}([0,T]), P \times \frac{1}{T}\text{Leb})$. Hence,

$$\lim_{\theta \to 0} E \frac{1}{T} \int_0^T \frac{1}{|\theta|} \left| \left( \mathcal{L}_\theta^C - \mathcal{L}_0^C \right) v_0(s, X_\theta^x(s)) - \theta^T \nabla_\theta \mathcal{L}_0^C v_0(s, X_0^x(s)) \right| ds$$

$$= E \frac{1}{T} \int_0^T \lim_{\theta \to 0} \frac{1}{|\theta|} \left| \left( \mathcal{L}_\theta^C - \mathcal{L}_0^C \right) v_0(s, X_\theta^x(s)) - \theta^T \nabla_\theta \mathcal{L}_0^C v_0(s, X_0^x(s)) \right| ds$$

We use the mean value theorem to get that for some $C > 0$ and $\xi_i = \xi_{\theta,i}(s, X_\theta^x(s)) \in (0,1), \eta_{i,j} = \eta_{\theta,i,j}(s, X_\theta^x(s)) \in (0,1)$,

$$\lim_{\theta \to 0} \frac{1}{|\theta|} \left| \left( \mathcal{L}_\theta^C - \mathcal{L}_0^C \right) v_0(s, X_\theta^x(s)) - \theta^T \nabla_\theta \mathcal{L}_0^C v_0(s, X_0^x(s)) \right|$$

$$\leq C \lim_{\theta \to 0} \left| \sum_{i=1}^d \nabla_\theta \mu_{\xi_i \theta, i}(s, X_\theta^x(s)) \partial_i v_0(s, X_\theta^x(s)) - \nabla_\theta \mu_{0,i}(s, X_0^x(s)) \partial_i v_0(s, X_0^x(s)) \right|$$

$$+ C \lim_{\theta \to 0} \sum_{i,j=1}^d \left| \nabla_\theta a_{\eta_{i,j}\theta, i, j}(s, X_\theta^x(s)) \partial_i \partial_j v_0(s, X_\theta^x(s)) - \nabla_\theta a_{0,i,j}(s, X_0^x(s)) \partial_i \partial_j v_0(s, X_0^x(s)) \right|$$

$$= 0$$

where the last equality follows from the continuity of $(\theta, x) \to \nabla_\theta \mu_{\theta,i}(s,x)$ and $\nabla_\theta a_{\theta,i,j}(s,x)$, $x \to \partial_i v_0(s,x)$ (Assumption 6) and $\partial_i \partial_j v_0(s,x)$ (Assumption 8), and $\theta \to X_\theta^x(s)$ (Theorem K).

Therefore, going back to the limit ratio in (B.6), we have shown that

$$\lim_{\theta \to 0} \frac{1}{|\theta|} \left| E \left[ \int_0^T \left( \mathcal{L}_\theta^C - \mathcal{L}_0^C \right) v_0(s, X_\theta^x(s)) ds \right] - \theta^T E \left[ \int_0^T \nabla_\theta \mathcal{L}_0^C v_0(s, X_0^x(s)) ds \right] \right| = 0$$

**The Jump Part:** Similar to the continuous part, we claim that the derivative should be

$$E \int_0^T \nabla_\theta \mathcal{L}_0^J v_0(s, X_0^x(s)) ds \tag{B.15}$$

where $\nabla_\theta \mathcal{L}_0^J$ is defined in (3.4).

To simplify notation, write

$$\left( \mathcal{L}_\theta^J - \mathcal{L}_0^J \right) v_0(s, X_\theta^x(s)) = \int_{\mathbb{R}_0^{d'}} D_1 - D_2 \nu(dz)$$

where

$$D_1 := v_0(s, X_\theta^x(s) + \chi_\theta(s, X_\theta^x(s), z)) - v_0(s, X_\theta^x(s) + \chi_0(s, X_\theta^x(s), z))$$

$$D_2 := \sum_{i=1}^d \left[ \chi_{\theta,i}(s, X_\theta^x(s), z) - \chi_{0,i}(s, X_\theta^x(s), z) \right] \partial_i v_0(s, X_\theta^x(s)).$$

Further, we write $\chi_\theta := \chi_\theta(s, X_\theta^x(s), z)$ and $\chi_{\theta,i} := \chi_{\theta,i}(s, X_\theta^x(s), z)$ when there is no ambiguity in the dependence on $s, X_\theta^x(s), z$. Then, apply the mean value theorem to $\rho \to v_0(s, X_\theta^x(s) + \rho\chi_\theta + (1-\rho)\chi_0)$, there exists $\xi = \xi_\theta(s, X_\theta^x(s), z) \in (0,1)$ s.t.

$$D_1 = \sum_{i=1}^d \left[ \chi_{\theta,i} - \chi_{0,i} \right] \partial_i v_0(s, X_\theta^x(s) + \xi\chi_\theta + (1-\xi)\chi_0),$$

Therefore,

$$D_1 - D_2 = \sum_{i=1}^d \left[ \chi_{\theta,i} - \chi_{0,i} \right] \left[ \partial_i v_0(s, X_\theta^x(s) + \xi\chi_\theta + (1-\xi)\chi_0) - \partial_i v_0(s, X_\theta^x(s)) \right]. \tag{B.16}$$

Again, we consider the limit

$$
\lim_{\theta \to 0} \frac{1}{|\theta|} \left| E\left[ \int_0^T \left( \mathcal{L}_\theta^J - \mathcal{L}_0^J \right) v_0(s, X_\theta^x(s)) ds \right] - \theta^T E\left[ \int_0^T \nabla_\theta \mathcal{L}_0^J v_0(s, X_0^x(s)) ds \right] \right|
$$

$$
\leq \lim_{\theta \to 0} E \int_0^T \frac{1}{|\theta|} \left| \left( \mathcal{L}_\theta^J - \mathcal{L}_0^J \right) v_0(s, X_\theta^x(s)) - \theta^T \nabla_\theta \mathcal{L}_0^J v_0(s, X_0^x(s)) \right| ds
$$

$$
\leq \lim_{\theta \to 0} E \int_0^T \int_{\mathbb{R}_0^{d'}} \frac{1}{|\theta| \gamma(z)^2} \left| D_1 - D_2 - \sum_{i=1}^d \theta^T \nabla_\theta \chi_{0,i} \left( \partial_i v_0(t, x + \chi_0) - \partial_i v_0(t, x) \right) \right| \mu(dz) ds
$$
(B.17)

where, as we will show below, the two pre-limit expectations in the first line are finite.

As before, we proceed show that the limit in $\theta$ can be exchanged into the triple integration by showing U.I. of

$$
\frac{1}{|\theta| \gamma(z)^2} \left| D_1 - D_2 - \sum_{i=1}^d \theta^T \nabla_\theta \chi_{0,i}(s, X_0^s(s), z) \left( \partial_i v_0(s, X_0^s(s) + \chi_0) - \partial_i v_0(s, X_0^s(s)) \right) \right|.
$$
(B.18)

on $\Omega \times [0, T] \times \mathbb{R}_0^{d'}$ w.r.t. the probability measure $P \times \frac{1}{T}\text{Leb} \times \frac{1}{\mu(\mathbb{R}_0^{d'})}\mu$. We consider

$$
E \frac{1}{T} \int_0^T \frac{1}{\mu(\mathbb{R}_0^{d'})} \int_{\mathbb{R}_0^{d'}} \frac{1}{|\theta|^\alpha \gamma(z)^{2\alpha}} \left| D_1 - D_2 - \sum_{i=1}^d \theta^T \nabla_\theta \chi_{0,i}(s, X_0^s(s), z) \left( \partial_i v_0(s, X_0^s(s) + \chi_0) - \partial_i v_0(s, X_0^s(s)) \right) \right|^\alpha \mu(dz) ds
$$

$$
\leq E \frac{1}{T} \int_0^T \frac{1}{\mu(\mathbb{R}_0^{d'})} \int_{\mathbb{R}_0^{d'}} \frac{|D_1 - D_2|^\alpha}{|\theta|^\alpha \gamma(z)^{2\alpha}} \mu(dz) ds
$$

$$
+ E \frac{1}{T} \int_0^T \frac{1}{\mu(\mathbb{R}_0^{d'})} \int_{\mathbb{R}_0^{d'}} \frac{1}{\gamma(z)^{2\alpha}} \left| \sum_{i=1}^d \nabla_\theta \chi_{0,i}(s, X_0^s(s), z) \left( \partial_i v_0(s, X_0^s(s) + \chi_0) - \partial_i v_0(s, X_0^s(s)) \right) \right|^\alpha \mu(dz) ds
$$

$$
=: E_1 + E_2
$$
(B.19)

We consider the two terms separately. For $E_1$, applying the mean value theorem again to (B.16), there exists $\eta_i = \eta_{\theta,i}(s, X_\theta^x(s), z, \xi)$ s.t.

$$
D_1 - D_2 = \sum_{i,j=1}^d [\chi_{\theta,i} - \chi_{0,i}] [\xi \chi_{\theta,j} + (1 - \xi)\chi_{0,j}] \partial_j \partial_i v_0(s, X_\theta^x(s) + \eta_i \xi \chi_\theta + \eta_i (1 - \xi)\chi_0)
$$

Therefore,

$$
\int_{\mathbb{R}_0^{d'}} \left| \frac{D_1 - D_2}{\gamma(z)^2} \right|^\alpha \mu(dz)
$$

$$
\leq d^{2(\alpha-1)} \int_{\mathbb{R}_0^{d'}} \left( \sum_{i,j=1}^d |\partial_j \partial_i v_0(s, X_\theta^x(s) + \eta_i \xi \chi_\theta + \eta_i (1 - \xi)\chi_0)|^2 \right)^{\frac{\alpha}{2}} \frac{1}{\gamma(z)^{2\alpha}} \sqrt{\sum_{i,j=1}^d |\chi_{\theta,i} - \chi_{0,i}|^{2\alpha} |\xi \chi_{\theta,j} + (1 - \xi)\chi_{0,j}|^{2\alpha}} \mu(dz)
$$

$$
\leq C \left( \int_{\mathbb{R}_0^{d'}} \left( \sum_{i,j=1}^d |\partial_j \partial_i v_0(s, X_\theta^x(s) + \eta_i \xi \chi_\theta + \eta_i (1 - \xi)\chi_0)|^2 \right)^\alpha \mu(dz) \sum_{i,j=1}^d \int_{\mathbb{R}_0^{d'}} \frac{|\chi_{\theta,i} - \chi_{0,i}|^{2\alpha} |\xi \chi_{\theta,j} + (1 - \xi)\chi_{0,j}|^{2\alpha}}{\gamma(z)^{4\alpha}} \mu(dz) \right)^{\frac{1}{2}}
$$

$$
=: C(I_1 \cdot I_2)^{1/2}
$$

We look at $I_1$ and $I_2$ separately. By Assumption 9,

$$\sum_{i=1}^{d}\sum_{j=1}^{d}|\partial_j\partial_i v_0(s, X_\theta^x(s) + \eta_i\xi\chi_\theta + \eta_i(1-\xi)\chi_0)|^2$$

$$\leq \sum_{i=1}^{d}|H[v_0](s, X_\theta^x(s) + \eta_i\xi\chi_\theta + \eta_i(1-\xi)\chi_0)|^2$$

$$\leq \sum_{i=1}^{d}c_v^2\left(|X_\theta^x(s) + \eta_i\xi\chi_\theta + \eta_i(1-\xi)\chi_0| + 1\right)^{2m}$$

$$\leq dc_v^2\left(|X_\theta^x(s)| + |\chi_\theta| + |\chi_0| + 1\right)^{2m}$$

$$\leq C\left(|X_\theta^x(s)|^{2m} + \frac{|\chi_\theta|^{2m}}{\gamma(z)^{2m}} + \frac{|\chi_0|^{2m}}{\gamma(z)^{2m}} + 1\right).$$

where we recall that $\gamma(z) = |z| \wedge 1 \leq 1$. Then, we consider, by Assumption 5, for $p \geq 2$

$$\chi_{p,\vee}^p := \sup_{\theta'\in\Theta, s\in[0,T]} E\int_{\mathbb{R}_0^{d'}} \frac{|\chi_{\theta'}(s, 0, z)|^p}{\gamma(z)^p}\mu(dz) < \infty \tag{B.20}$$

and for all $\theta, \theta' \in \Theta$

$$\int_{\mathbb{R}_0^{d'}} \frac{|\chi_{\theta'}(s, X_\theta^x(s), z) - \chi_{\theta'}(s, 0, z)|^p}{\gamma(z)^p}\mu(dz) \leq c_p^p|X_\theta^x(s)|^p.$$

So, for $p \geq 2$

$$\int_{\mathbb{R}_0^{d'}} \frac{|\chi_{\theta'}(s, X_\theta^x(s), z)|^p}{\gamma(z)^p}\mu(dz) \leq C|X_\theta^x(s)|^p + \int_{\mathbb{R}_0^{d'}} \frac{|\chi_{\theta'}(s, 0, z)|^p}{\gamma(z)^p}\mu(dz)$$

$$\leq C|X_\theta^x(s)|^p + \chi_{p,\vee}^p. \tag{B.21}$$

As $\mu$ is a finite measure, we have

$$I_1 = \int_{\mathbb{R}_0^{d'}} \left(C\left(|X_\theta^x(s)|^{2m} + \frac{|\chi_\theta|^{2m}}{\gamma(z)^{2m}} + \frac{|\chi_0|^{2m}}{\gamma(z)^{2m}} + 1\right)\right)^\alpha \mu(dz)$$

$$\leq C(|X_\theta^x(s)| + 1)^{2\alpha m}.$$

For $I_2$, we bound

$$I_2 = \sum_{i,j=1}^{d}\int_{\mathbb{R}_0^{d'}} \frac{|\chi_{\theta,i} - \chi_{0,i}|^{2\alpha}|\xi\chi_{\theta,j} + (1-\xi)\chi_{0,j}|^{2\alpha}}{\gamma(z)^{4\alpha}}\mu(dz)$$

$$\leq \sum_{i,j=1}^{d}\left(\int_{\mathbb{R}_0^{d'}} \frac{|\xi\chi_{\theta,j} + (1-\xi)\chi_{0,j}|^{4\alpha}}{\gamma(z)^{4\alpha}}\mu(dz)\int_{\mathbb{R}_0^{d'}} \frac{|\chi_{\theta,i} - \chi_{0,i}|^{4\alpha}}{\gamma(z)^{4\alpha}}\mu(dz)\right)^{1/2}$$

$$\leq d\left(\sum_{j=1}^{d}\int_{\mathbb{R}_0^{d'}} \frac{|\xi\chi_{\theta,j} + (1-\xi)\chi_{0,j}|^{4\alpha}}{\gamma(z)^{4\alpha}}\mu(dz)\sum_{i=1}^{d}\int_{\mathbb{R}_0^{d'}} \frac{|\chi_{\theta,i} - \chi_{0,i}|^{4\alpha}}{\gamma(z)^{4\alpha}}\mu(dz)\right)^{1/2}$$

$$\leq d2^{4\alpha-1}\left(\sum_{j=1}^{d}\int_{\mathbb{R}_0^{d'}} \frac{|\chi_{\theta,j}|^{4\alpha} + |\chi_{0,j}|^{4\alpha}}{\gamma(z)^{4\alpha}}\mu(dz)\sum_{i=1}^{d}\int_{\mathbb{R}_0^{d'}} \frac{|\chi_{\theta,i} - \chi_{0,i}|^{4\alpha}}{\gamma(z)^{4\alpha}}\mu(dz)\right)^{1/2}$$

$$\leq C\left(\int_{\mathbb{R}_0^{d'}} \frac{|\chi_\theta|^{4\alpha} + |\chi_0|^{4\alpha}}{\gamma(z)^{4\alpha}}\mu(dz)\int_{\mathbb{R}_0^{d'}} \frac{|\chi_\theta - \chi_0|^{4\alpha}}{\gamma(z)^{4\alpha}}\mu(dz)\right)^{1/2}$$

By Assumption 5,

$$\int_{\mathbb{R}_0^{d'}} \frac{|\chi_\theta - \chi_0|^{4\alpha}}{\gamma(z)^{4\alpha}} \mu(dz) \le \kappa_{\theta,0}^{4\alpha}(s)(|X_\theta^x(s)| + 1)^{4\alpha}.$$

Use this and inequality (B.21), we obtain

$$\begin{aligned}
I_2 &= \sum_{i,j=1}^{d} \int_{\mathbb{R}_0^{d'}} \frac{|\chi_{\theta,i} - \chi_{0,i}|^{2\alpha} |\xi\chi_{\theta,j} + (1-\xi)\chi_{0,j}|^{2\alpha}}{\gamma(z)^{4\alpha}} \mu(dz) \\
&\le C \left[ 2 \left( C|X_\theta^x(s)|^{4\alpha} + \chi_{4\alpha,\vee}^{4\alpha} \right) \kappa_{\theta,0}^{4\alpha}(s)(|X_\theta^x(s)| + 1)^{4\alpha} \right]^{1/2} \\
&\le C\kappa_{\theta,0}^{4\alpha}(s)^{1/2}(|X_\theta^x(s)| + 1)^{4\alpha}
\end{aligned}$$

In summary, we have

$$\begin{aligned}
\int_{\mathbb{R}_0^{d'}} \left| \frac{D_1 - D_2}{\gamma(z)^2} \right|^\alpha \mu(dz) &\le C(I_1 \cdot I_2)^{1/2} \\
&\le C\kappa_{\theta,0}^{4\alpha}(s)^{1/4}(|X_\theta^x(s)| + 1)^{(m+2)\alpha}.
\end{aligned}$$

Therefore, by the same argument as in the derivation (B.9), we conclude that

$$\begin{aligned}
\sup_{\theta \in \Theta} E_1 &= \sup_{\theta \in \Theta} E \frac{1}{T} \int_0^T \frac{1}{\mu(\mathbb{R}_0^{d'})} \int_{\mathbb{R}_0^{d'}} \frac{|D_1 - D_2|^\alpha}{|\theta|^\alpha \gamma(z)^{2\alpha}} \mu(dz) ds \\
&\le C \sup_{\theta \in \Theta} \frac{1}{|\theta|^\alpha} E \int_0^T \kappa_{\theta,0}^{4\alpha}(s)^{1/4}(|X_\theta^x(s)| + 1)^{(m+2)\alpha} ds \\
&\le C \sup_{\theta \in \Theta} \frac{1}{|\theta|^\alpha} \int_0^T \kappa_{\theta,0}^{4\alpha}(s)^{1/4} ds \cdot \sup_{\theta \in \Theta, s \in [0,T]} E(|X_\theta^x(s)| + 1)^{(m+2)\alpha} \quad \text{(B.22)} \\
&\overset{(i)}{\le} C \sup_{\theta \in \Theta} \frac{1}{|\theta|^\alpha} \left( \int_0^T \kappa_{\theta,0}^{4\alpha}(s) ds \right)^{1/4} \left( b_{(m+2)\alpha}^{(m+2)\alpha}(|x| + 1)^{(m+2)\alpha} + 1 \right) \\
&\le C(|x| + 1)^{(m+2)\alpha}.
\end{aligned}$$

where $(i)$ uses Jensen's inequality and Theorem K. Note that, with $\alpha = 1$, this also implies the finiteness of the first expectation in (B.17).

For the second term in (B.19), we use the same technique as in the derivation for that of the continuous part. First, we consider

$$\int_{\mathbb{R}_0^{d'}} \frac{1}{\gamma(z)^{2\alpha}} \left| \sum_{i=1}^{d} \nabla_\theta \chi_{0,i}(s,x,z) \left( \partial_i v_0(s, x + \chi_0(s,x,z)) - \partial_i v_0(s,x) \right) \right|^\alpha \mu(dz)$$

$$\leq C \int_{\mathbb{R}_0^{d'}} \frac{1}{\gamma(z)^{2\alpha}} \sum_{i=1}^{d} \left| \nabla_\theta \chi_{0,i}(s,x,z) \left( \partial_i v_0(s, x + \chi_0(s,x,z)) - \partial_i v_0(s,x) \right) \right|^\alpha \mu(dz)$$

$$\leq C \int_{\mathbb{R}_0^{d'}} \frac{1}{\gamma(z)^{2\alpha}} \sum_{i=1}^{d} \left| \partial_i v_0(s, x + \chi_0(s,x,z)) - \partial_i v_0(s,x) \right|^\alpha \left| \nabla_\theta \chi_{0,i}(s,x,z) \right|^\alpha \mu(dz)$$

$$\overset{(i)}{=} C \int_{\mathbb{R}_0^{d'}} \frac{1}{\gamma(z)^{2\alpha}} \sum_{i=1}^{d} \left| \sum_{j=1}^{d} \partial_j \partial_i v_0(s, x + \xi_i \chi_0(s,x,z)) \chi_{0,j}(s,x,z) \right|^\alpha \sum_{l=1}^{n} \left| \partial_{\theta_l} \chi_{0,i}(s,x,z) \right|^\alpha \mu(dz)$$

$$\leq C \int_{\mathbb{R}_0^{d'}} \frac{1}{\gamma(z)^{2\alpha}} \sum_{i=1}^{d} \left| \chi_0(s,x,z) \right|^\alpha \left| \sum_{j=1}^{d} \left| \partial_j \partial_i v_0(s, x + \xi_i \chi_0(s,x,z)) \right|^2 \right|^{\frac{\alpha}{2}} \sum_{l=1}^{n} \left| \partial_{\theta_l} \chi_{0,i}(s,x,z) \right|^\alpha \mu(dz)$$

$$\leq C \int_{\mathbb{R}_0^{d'}} \frac{1}{\gamma(z)^{2\alpha}} \sum_{i=1}^{d} \left| \chi_0(s,x,z) \right|^\alpha \left| H[v_0](s, x + \xi_i \chi_0(s,x,z)) \right|^\alpha \sum_{l=1}^{n} \left| \partial_{\theta_l} \chi_{0,i}(s,x,z) \right|^\alpha \mu(dz)$$

$$\overset{(ii)}{\leq} C \int_{\mathbb{R}^{d'}} \frac{\left| \chi_0(s,x,z) \right|^\alpha}{\gamma(z)^{2\alpha}} \sum_{i=1}^{d} (|x + \xi_i \chi_0(s,x,z)| + 1)^{m\alpha} \sum_{l=1}^{n} \left| \partial_{\theta_l} \chi_{0,i}(s,x,z) \right|^\alpha \mu(dz)$$

$$\overset{(iii)}{\leq} C \int_{\mathbb{R}_0^{d'}} \frac{\left| \chi_0(s,x,z) \right|^\alpha}{\gamma(z)^{2\alpha}} \left[ (|x| + 1)^{m\alpha} + \frac{\left| \chi_0(s,x,z) \right|^{m\alpha}}{\gamma(z)^{m\alpha}} \right] \sum_{i=1}^{d} \sum_{l=1}^{n} \left| \partial_{\theta_l} \chi_{0,i}(s,x,z) \right|^\alpha \mu(dz)$$

$$\leq C(|x| + 1)^{m\alpha} \sum_{i=1}^{d} \sum_{l=1}^{n} \int_{\mathbb{R}_0^{d'}} \frac{\left| \chi_0(s,x,z) \right|^\alpha}{\gamma(z)^\alpha} \frac{\left| \partial_{\theta_l} \chi_{0,i}(s,x,z) \right|^\alpha}{\gamma(z)^\alpha} \mu(dz)$$

$$+ C \sum_{i=1}^{d} \sum_{l=1}^{n} \int_{\mathbb{R}_0^{d'}} \frac{\left| \chi_0(s,x,z) \right|^{(m+1)\alpha}}{\gamma(z)^{(m+1)\alpha}} \frac{\left| \partial_{\theta_l} \chi_{0,i}(s,x,z) \right|^\alpha}{\gamma(z)^\alpha} \mu(dz)$$

$$\leq C(|x| + 1)^{m\alpha} \sum_{i=1}^{d} \sum_{l=1}^{n} \left( \int_{\mathbb{R}_0^{d'}} \frac{\left| \chi_0(s,x,z) \right|^{2\alpha}}{\gamma(z)^{2\alpha}} \mu(dz) \int_{\mathbb{R}_0^{d'}} \frac{\left| \partial_{\theta_l} \chi_{0,i}(s,x,z) \right|^{2\alpha}}{\gamma(z)^{2\alpha}} \mu(dz) \right)^{\frac{1}{2}}$$

$$+ C \sum_{i=1}^{d} \sum_{l=1}^{n} \left( \int_{\mathbb{R}_0^{d'}} \frac{\left| \chi_0(s,x,z) \right|^{2(m+1)\alpha}}{\gamma(z)^{2(m+1)\alpha}} \mu(dz) \int_{\mathbb{R}_0^{d'}} \frac{\left| \partial_{\theta_l} \chi_{0,i}(s,x,z) \right|^{2\alpha}}{\gamma(z)^{2\alpha}} \mu(dz) \right)^{\frac{1}{2}}$$

$$\overset{(iv)}{\leq} C(|x| + 1)^{(m+1)\alpha} \sum_{i=1}^{d} \sum_{l=1}^{n} \left( \int_{\mathbb{R}_0^{d'}} \frac{\left| \partial_{\theta_l} \chi_{0,i}(s,x,z) \right|^{2\alpha}}{\gamma(z)^{2\alpha}} \mu(dz) \right)^{\frac{1}{2}}$$

where $(i)$ applies the mean value theorem to $\rho \to \partial_i v_0(s, x + \rho \chi_0)$ to yield the existence of such $\xi_i := \xi_i(s,x,z)$, $(ii)$ follows from Assumption 9, and $(iii)$ uses Hölder's inequality $\|fg\|_1 \leq \|f\|_\infty \|g\|_1$ as well as $\gamma(z) \leq 1$, and $(iv)$ follows from (B.21) where we have that for $p \geq 2$

$$\int_{\mathbb{R}_0^{d'}} \frac{\left| \chi_0(s,x,z) \right|^p}{\gamma(z)^p} \mu(dz) \leq C(|x| + 1)^p.$$

Therefore, by Theorem K and Cauchy-Schwarz inequality,

$$
\begin{aligned}
E_2 &= E\frac{1}{T}\int_0^T \frac{1}{\mu(\mathbb{R}_0^{d'})}\int_{\mathbb{R}_0^{d'}} \frac{1}{\gamma(z)^{2\alpha}} \left|\sum_{i=1}^d \nabla_\theta\chi_{0,i}(s,X_0^s(s),z)\left(\partial_i v_0(s,X_0^s(s)+\chi_0)-\partial_i v_0(s,X_0^s(s))\right)\right|^\alpha \mu(dz) \\
&\leq C\sum_{i=1}^d\sum_{l=1}^n E\frac{1}{T}\int_0^T (|X_0^x(s)|+1)^{(m+1)\alpha}\left(\int_{\mathbb{R}_0^{d'}} \frac{|\partial_{\theta_l}\chi_{0,i}(s,X_0^x(s),z)|^{2\alpha}}{\gamma(z)^{2\alpha}}\mu(dz)\right)^{1/2}ds \\
&\leq Cb_{2(m+1)\alpha}^{(m+1)\alpha}(|x|+1)^{(m+1)\alpha}\sum_{i=1}^d\sum_{l=1}^n\left(E\frac{1}{T}\int_0^T\int_{\mathbb{R}_0^{d'}} \frac{|\partial_{\theta_l}\chi_{0,i}(s,X_0^x(s),z)|^{2\alpha}}{\gamma(z)^{2\alpha}}\mu(dz)ds\right)^{1/2}
\end{aligned}
$$

To bound this, as in the continuous part, we check the uniform integrability on $\Omega\times[0,T]\times\mathbb{R}_0^{d'}$ w.r.t. the probability measure $P\times\frac{1}{T}\mathrm{Leb}\times\frac{1}{\mu(\mathbb{R}_0^{d'})}\mu$ when $\delta$ is in a sufficiently small neighbourhood of $0$ of the derivative ratio

$$
\frac{1}{\gamma(z)^{2\alpha}}\left(\frac{|\chi_{\delta e_l,i}(s,X_0^x(s),z)-\chi_{0,i}(s,X_0^x(s),z)|}{\delta}\right)^{2\alpha}. \tag{B.23}
$$

To simplify notation, we again denote $\chi_{\theta,i}:=\chi_{\theta,i}(s,X_0^x(s),z)$. To check this, we consider for $\epsilon\geq 0$

$$
\begin{aligned}
&E\frac{1}{T}\int_0^T\frac{1}{\mu(\mathbb{R}_0^{d'})}\int_{\mathbb{R}_0^{d'}}\frac{1}{\gamma(z)^{2\alpha+\epsilon}}\left(\frac{|\chi_{\delta e_l,i}-\chi_{0,i}|}{\delta}\right)^{2\alpha+\epsilon}\mu(dz)ds \\
&= \frac{1}{\mu(\mathbb{R}_0^{d'})}E\frac{1}{T}\int_0^T\frac{1}{\delta^{2\alpha+\epsilon}}\int_{\mathbb{R}_0^{d'}}\left(\frac{|\chi_{\delta e_l,i}-\chi_{0,i}|}{\gamma(z)}\right)^{2\alpha+\epsilon}\mu(dz)ds \\
&\leq \frac{1}{\mu(\mathbb{R}_0^{d'})}\frac{1}{\delta^{2\alpha+\epsilon}}E\frac{1}{T}\int_0^T \kappa_{\delta e_l,0}^{2\alpha+\epsilon}(s)(|X_0^x(s)|+1)^{2\alpha+\epsilon}ds \\
&\leq C\frac{1}{\delta^{2\alpha+\epsilon}}\int_0^T \kappa_{\delta e_l,0}^{2\alpha+\epsilon}(s)ds\cdot E\sup_{s\in[0,T]}(|X_0^x(s)|+1)^{2\alpha+\epsilon} \\
&\leq Cl_{2\alpha+\epsilon}^{2\alpha+\epsilon}b_{2\alpha+\epsilon}^{2\alpha+\epsilon}((|x|+1)^{2\alpha+\epsilon}+1)
\end{aligned}
$$

independent of $\delta$. Choose $\epsilon>0$ will show the U.I. of (B.23). Therefore, we

$$
\begin{aligned}
E_2 &\leq C(|x|+1)^{(m+1)\alpha}\sum_{i=1}^d\sum_{l=1}^n\left(E\frac{1}{T}\int_0^T\frac{1}{\mu(\mathbb{R}_0^{d'})}\int_{\mathbb{R}_0^{d'}}\frac{|\partial_{\theta_l}\chi_{0,i}(s,X_0^x(s),z)|^{2\alpha}}{\gamma(z)^{2\alpha}}\mu(dz)ds\right)^{1/2} \\
&= C(|x|+1)^{(m+1)\alpha}\sum_{i=1}^d\sum_{l=1}^n\left(\lim_{\delta\downarrow 0}E\frac{1}{T}\int_0^T\frac{1}{\mu(\mathbb{R}_0^{d'})}\int_{\mathbb{R}_0^{d'}}\frac{1}{\gamma(z)^{2\alpha}}\left(\frac{|\chi_{\delta e_l,i}-\chi_{0,i}|}{\delta}\right)^{2\alpha}\mu(dz)ds\right)^{1/2} \\
&\leq C(|x|+1)^{(m+2)\alpha}
\end{aligned}
$$

where the last inequality follows from previous derivation with $\epsilon=0$. In particular, recalling the definition of $E_2$ in (B.19), this shows that

$$
E\left|\int_0^T \nabla_\theta\mathcal{L}_0^J v_0(s,X_0^x(s))ds\right|<\infty
$$

as claimed above.

Therefore, by bounding the two terms in (B.19), we conclude the uniform integrability of (B.18). So, going back to (B.17), U.I. implies that

$$\lim_{\theta \to 0} \frac{1}{|\theta|} \left| E\left[ \int_0^T \left( \mathcal{L}_\theta^J - \mathcal{L}_0^J \right) v_0(s, X_\theta^x(s)) ds \right] - \theta^T E\left[ \int_0^T \nabla_\theta \mathcal{L}_0^J v_0(s, X_0^x(s)) ds \right] \right|$$

$$\leq \lim_{\theta \to 0} E \int_0^T \int_{\mathbb{R}_0^{d'}} \frac{1}{|\theta| \gamma(z)^2} \left| D_1 - D_2 - \sum_{i=1}^d \theta^T \nabla_\theta \chi_{0,i} \left( \partial_i v_0(s, X_0^x(s) + \chi_0) - \partial_i v_0(s, X_0^x(s)) \right) \right| \mu(dz) ds$$

$$= E \int_0^T \int_{\mathbb{R}_0^{d'}} \frac{1}{\gamma(z)^2} \lim_{\theta \to 0} \frac{1}{|\theta|} \left| D_1 - D_2 - \sum_{i=1}^d \theta^T \nabla_\theta \chi_{0,i} \left( \partial_i v_0(s, X_0^x(s) + \chi_0) - \partial_i v_0(s, X_0^x(s)) \right) \right| \mu(dz) ds$$

$$= 0,$$

where the last step follows from

$$\lim_{\theta \to 0} \frac{1}{|\theta|} \left[ D_1 - D_2 - \sum_{i=1}^d \theta^T |\theta| \nabla_\theta \chi_{0,i}(s, X_0^x(s), z) \left( \partial_i v_0(s, X_0^x(s) + \chi_0) - \partial_i v_0(s, X_0^x(s)) \right) \right]$$

$$= \sum_{i=1}^d \left( \partial_i v_0(s, X_0^x(s) + \chi_0) - \partial_i v_0(s, X_0^x(s)) \right) \lim_{\theta \to 0} \frac{1}{|\theta|} \left( \chi_{\theta,i}(s, X_\theta^x(s), z) - \chi_{0,i}(s, X_\theta^x(s), z) - \theta^T \nabla_\theta \chi_{0,i}(s, X_0^x(s), z) \right)$$

$$\overset{(i)}{=} \sum_{i=1}^d \left( \partial_i v_0(s, X_0^x(s) + \chi_0) - \partial_i v_0(s, X_0^x(s)) \right) \lim_{\theta \to 0} \frac{1}{|\theta|} \theta^T \left( \nabla_\theta \chi_{\xi_i \theta, i}(s, X_\theta^x(s), z) - \nabla_\theta \chi_{0,i}(s, X_0^x(s), z) \right)$$

$$\overset{(ii)}{=} 0.$$

Here, $(i)$ applies the mean value theorem and $(ii)$ use the continuity of $\theta \to \nabla_\theta \chi_{\xi_i \theta, i}(s, X_\theta^x(s), z)$ as in Assumption 6.

**The Rewards Part:** We first consider the reward rate $r$. As in the previous proof, we show the U.I. of

$$I_1(\theta) := \frac{1}{|\theta|^\alpha} E \frac{1}{T} \int_0^T |\rho_\theta(s, X_\theta^x(s)) - \rho_0(s, X_\theta^x(s))|^\alpha ds$$

and the finiteness of

$$I_2 := \frac{1}{|\theta|^\alpha} E \frac{1}{T} \int_0^T |\nabla_\theta \rho_0(s, X_\theta^x(s))|^\alpha ds$$

for some $\alpha > 1$. By Assumption 7 item 1 and the same derivation as in (B.9),

$$I_1(\theta) \leq \frac{C}{|\theta|^\alpha} E \frac{1}{T} \int_0^T \kappa_{\theta,0}^\alpha(s)(|X_\theta^x(s)| + 1)^{m\alpha} ds$$

$$\leq \frac{C}{|\theta|^\alpha} \int_0^T \kappa_{\theta,0}^\alpha(s) ds \sup_{\theta \in \Theta, s \in [0,T]} E(|X_\theta^x(s)| + 1)^{m\alpha}$$

$$\leq C(|x| + 1)^{m\alpha}$$

uniformly in $\theta$. Moreover,

$$E \frac{1}{T} \int_0^T |\partial_{\theta_l} \rho_0(s, X_0^x(s))|^\alpha ds = E \frac{1}{T} \int_0^T \lim_{\delta \downarrow 0} \left| \frac{r_{\delta e_l, i}(s, X_0^x(s)) - r_{0,i}(s, X_0^x(s))}{\delta} \right|^\alpha ds$$

$$= \lim_{\delta \downarrow 0} E \frac{1}{T} \int_0^T \left| \frac{r_{\delta e_l, i}(s, X_0^x(s)) - r_{0,i}(s, X_0^x(s))}{\delta} \right|^\alpha ds$$

$$\leq \sup_{\theta \in \Theta} \frac{1}{|\theta|^\alpha} E \frac{1}{T} \int_0^T \kappa_{\theta,0}^\alpha(s)(1 + |X_0^x(s)|)^\alpha ds$$

$$< \infty.$$

These results and the continuity of $(\theta, x) \to \nabla_\theta r(s, x)$ and $\theta \to X_\theta^x(s)$ implies that

$$\lim_{\theta \to 0} \frac{1}{|\theta|} \left| E\left[ \int_0^T \rho_\theta(s, X_\theta^x(s)) - \rho_0(s, X_\theta^x(s)) ds \right] - \theta^T E\left[ \int_0^T \nabla_\theta \rho_0(s, X_0^x(s)) ds \right] \right|$$

$$\leq E\left[ \int_0^T \lim_{\theta \to 0} \frac{1}{|\theta|} \left| \rho_\theta(s, X_\theta^x(s)) - \rho_0(s, X_\theta^x(s)) - \theta^T \nabla_\theta \rho_0(s, X_0^x(s)) \right| ds \right]$$

$$= 0.$$

For the terminal reward $g$ term, the same proof with the integral removed and $s$ replaced by $T$ will yield the desired conclusion.

## C  Proof of Proposition A.1

We note that the statement for $X_0^x$ and its first derivative holds from directly applying Kunita [11, Theorem 3.3.2] and the a.s. version of Kolmogorov's continuity criterion as in Corollary 1 of Protter [19, Theorem 73].

To show the statement for the second derivative, we apply the proof of Kunita [11, Theorem 3.4.2]. From display (3.43), we look at the random drift:

$$M_{a,b,i}^x(r, H_{a,b}) := \sum_{l=1}^d \left[ \partial_l \mu_{0,i}(r, X_0^x(s, r)) H_{a,b,l} + \sum_{m=1}^d \partial_m \partial_l \mu_{0,i}(r, X_0^x(s, r)) \partial_a X_{0,l}^x(s, r) \partial_b X_{0,m}^x(s, r) \right]$$

seen as a function of $r \in [0, T]$, $H \in \mathbb{R}^{d \times d \times d}$, and show that it satisfies the conditions for Kunita [11, Theorem 3.3.2] with $\lambda = x$; i.e. the conditions in Assumption 10.

We note that as $\mu_0$ and $\partial_m \mu_0(r, x)$ satisfying item 2 of Assumption 10 for any $l, m = 1, \ldots, d$,

$$\sup_{r \in [0,T]} |\partial_m \mu_0(r, x)| = \sup_{r \in [0,T]} \left| \lim_{\delta \downarrow 0} \frac{\mu_0(r, x + \delta e_m) - \mu_0(r, x)}{\delta} \right|$$

$$\leq \sup_{r \in [0,T]} \lim_{\delta \downarrow 0} \frac{|\mu_0(r, x + \delta e_m) - \mu_0(r, x)|}{\delta}$$

$$\leq c$$

is bounded. Same for $\partial_m \partial_l \mu_0$.

First, at $H_{a,b} = 0$.

$$\sup_{r \in [0,T], x \in \mathbb{R}^d} E|M_{a,b,\cdot}^x(r, 0)|^p \leq C \sup_{r \in [0,T], x \in \mathbb{R}} \sum_{m,l=1}^d E|\partial_a X_{0,l}^x(s, r)||\partial_b X_{0,m}^x(s, r)| < \infty.$$

Second, $M_{a,b,i}^x(r, H_{a,b})$ is clearly uniformly Lipschitz in $H_{a,b}$ as $\partial_l \mu_0$ is bounded.

Third, using the boundedness of $\partial_m \mu_0$ and $\partial_m \partial_l \mu_0$, we have

$$\sum_{l=1}^d |\partial_l \mu_{0,i}(r, X_0^x(s, r)) H_{a,b,l} - \partial_l \mu_{0,i}(r, X_0^y(s, r)) H_{a,b,l}| \leq C|X_0^x(s, r) - X_0^y(s, r)||H_{a,b}|$$

and

$$\sum_{m,l=1}^{d} \left| \partial_m \partial_l \mu_{0,i}(r, X_0^x(s,r)) \partial_a X_{0,l}^x(s,r) \partial_b X_{0,m}^x(s,r) - \partial_m \partial_l \mu_{0,i}(r, X_0^y(s,r)) \partial_a X_{0,l}^y(s,r) \partial_b X_{0,m}^y(s,r) \right|$$

$$\leq \sum_{m,l=1}^{d} |\partial_m \partial_l \mu_{0,i}(r, X_0^x(s,r)) - \partial_m \partial_l \mu_{0,i}(r, X_0^y(s,r))| \left| \partial_a X_{0,l}^y(s,r) \partial_b X_{0,m}^y(s,r) \right|$$

$$+ |\partial_m \partial_l \mu_{0,i}(r, X_0^x(s,r))| \left| \partial_a X_{0,l}^x(s,r) \partial_b X_{0,m}^x(s,r) - \partial_a X_{0,l}^y(s,r) \partial_b X_{0,m}^y(s,r) \right|$$

$$\leq \sum_{m,l=1}^{d} C|X_0^x(s,r) - X_0^y(s,r)| \left| \partial_a X_{0,l}^y(s,r) \partial_b X_{0,m}^y(s,r) \right| + C \left| \partial_a X_{0,l}^x(s,r) - \partial_a X_{0,l}^y(s,r) \right| \left| \partial_b X_{0,m}^x(s,r) \right|$$

$$+ C \left| \partial_b X_{0,l}^x(s,r) - \partial_b X_{0,l}^y(s,r) \right| \left| \partial_a X_{0,m}^x(s,r) \right|.$$

Therefore, defining $K_{x,y}(r,H)$ to be the sum of the two, we see that

$$E \int_0^T K_{x,y}^{(a,b)}(r,H)^p dr \leq C|H_{a,b}|^p |x-y|^p + C|x-y|^p \leq C|x-y|^p(|H_{a,b}|+1)^p.$$

Here the first inequality follows from the first derivative satisfying the Proposition A.1, which follows from a direct application of Kunita [11, Theorem 3.3.2].

Similar results can be established for the volatility and the jump coefficients. Therefore, we conclude the proof by applying Kunita [11, Theorem 3.3.2] to the derivative and the second derivatives.

## D  Proof of Theorem 2'

Our proof of Theorem 2' hinges on the ability to exchange the derivative with the expectation and time integral. To achieve this, first, we use similar techniques as in the proof of Theorem 1' to prove the following lemma.

**Lemma 2.** *Under the assumptions of Theorem 2', for $h(t,x) = \rho_0(t,x)$ and $g_0(x)$, we have*

$$\partial_{x_i} E h(s, X_0^x(t,s)) = E \partial_{x_i} h(s, X_0^x(t,s)) = E \nabla h(s, X_0^x(t,s))^\top \partial_i X_0^x(t,s) \qquad \text{(D.1)}$$

*and*

$$\partial_{x_j} \partial_{x_i} E h(s, X_0^x(t,s)) = E \partial_{x_j} \partial_{x_i} h(s, X_0^x(t,s))$$
$$= E \partial_i X_0^x(t,s)^\top H[h](s, X_0^x(t,s)) \partial_j X_0^x(t,s) + \nabla h(s, X_0^x(t,s))^\top \partial_j \partial_i X_0^x(t,s)). \qquad \text{(D.2)}$$

*Moreover, there exists a constant $C$ independent of $t, s$ s.t.*

$$E|\partial_{x_i} h(s, X_0^x(t,s))| \leq C(|x|+1)^m, \quad \text{and} \quad E|\partial_{x_j} \partial_{x_i} h(s, X_0^x(t,s))| \leq C(|x|+1)^m.$$

Lemma 2 directly implies that the derivatives of the expected terminal rewards in (2.6) satisfy

$$\nabla_x E g_0^\top(t, X_0^x(t,T)) = E \nabla g_0^\top \nabla X_0^x(t,T),$$
$$H_x[E g_0^\top(t, X_0^x(t,T))] = E \left[ \nabla X_0^x(t,T)^\top H[g_0] \nabla X_0^x(t,T) + \left\langle \nabla g_0, H[X_{0,\cdot}^x](t,T) \right\rangle \right]. \qquad \text{(D.3)}$$

By the same argument, to prove Theorem 2', it suffices to show that for the cumulative reward parts in (2.6), the time integral and space derivatives can be interchanged. First, by Lemma 2, we see that

$$\int_t^T E|\partial_{x_i} \rho_0(s, X_0^x(t,s))| ds < \infty, \quad \text{and} \quad \int_t^T E|\partial_{x_j} \partial_{x_i} \rho_0(s, X_0^x(t,s))| ds < \infty.$$

So, by Fubini's theorem and Lemma 2

$$E\int_t^T \partial_{x_i}\rho_0(s,X_0^x(t,s))ds = \int_t^T E\partial_{x_i}\rho_0(s,X_0^x(t,s))ds$$

$$= \int_t^T \partial_{x_i}E\rho_0(s,X_0^x(t,s))ds$$

$$\stackrel{(i)}{=} \partial_{x_i}\int_t^T E\rho_0(s,X_0^x(t,s))ds$$

$$= \partial_{x_i}E\int_t^T \rho_0(s,X_0^x(t,s))ds$$

where $(i)$ follows from dominated convergence that for $y$ in a $\epsilon$ neighbourhood of $x$,

$$|\partial_{y_i}E\rho_0(s,X_0^y(t,s))| \le E|\partial_{y_i}\rho_0(s,X_0^y(t,s))| \le C(|x|+\epsilon+1)^m$$

independent of $s$. Similarly,

$$E\int_t^T \partial_{x_j}\partial_{x_i}\rho_0(s,X_0^x(t,s))ds = \partial_{x_j}\partial_{x_i}E\int_t^T \rho_0(s,X_0^x(t,s))ds.$$

This and (D.3) implies (2.6), completing the proof.

## D.1  Proof of Lemma 2

**First Space Derivatives:** We first show equality (D.1). Consider

$$\partial_{x_i}Eh(s,X_0^x(t,s)) = \lim_{\delta\to 0}\frac{1}{\delta}E\left[h(s,X_0^{x+\delta e_i}(t,s)) - h(s,X_0^x(t,s))\right]. \tag{D.4}$$

We exchange the limit and the expectation by considering

$$E\delta^{-\alpha}\left|h(s,X_0^{x+\delta e_i}(t,s)) - h(s,X_0^x(t,s))\right|^\alpha$$

$$= E\delta^{-\alpha}\left|\nabla h(s,\xi X_0^{x+\delta e_j}(t,s) + (1-\xi)X_0^x(t,s))^\top\left(X_0^{x+\delta e_j}(t,s) - X_0^x(t,s)\right)\right|^\alpha$$

$$\le \left(E\left|\frac{X_0^{x+\delta e_j}(t,s) - X_0^x(t,s)}{\delta}\right|^{2\alpha} E|\nabla h(s,\xi X_0^{x+\delta e_j}(t,s) + (1-\xi)X_0^x(t,s))|^{2\alpha}\right)^{1/2}$$

where the mean value theorem implies the existence of such r.v. $\xi \in [0,1]$. For the first term, Proposition A.1 implies that

$$E\left|\frac{X_0^{x+\delta e_j}(t,s) - X_0^x(t,s)}{\delta}\right|^{2\alpha} \le l_{2\alpha}^{2\alpha}.$$

For the second term, by Assumption 4

$$E|\nabla h(s,\xi X_0^{x+\delta e_j}(t,s) + (1-\xi)X_0^x(t,s))|^{2\alpha}$$
$$\le c_h^{2\alpha}E(|X_0^{x+\delta e_i}(t,s)| + |X_0^x(t,s)| + 1)^{2\alpha m}$$
$$\le c_h^{2\alpha}E(|X_0^{x+\delta e_i}(t,s) - X_0^x(t,s)| + 2|X_0^x(t,s)| + 1)^{2\alpha m} \tag{D.5}$$
$$\le C(|x|+1)^{2\alpha m} + Cl_{2\alpha m}^{2\alpha m}|\delta|^{2\alpha m}$$
$$\le C(|x|+1)^{2\alpha m}.$$

where the last inequality considers $|\delta| \le 1$ and $C$ can be chosen so that it doesn't depend on $\delta$, $s$, and $t$. Therefore, the limit in the r.h.s. of (D.4) can be interchanged with the expectation and we have that

$$\partial_{x_i}Eh(s,X_0^x(t,s)) = E\partial_{x_i}h(s,X_0^x(t,s))$$
$$= E\nabla h(s,X_0^x(t,s))^\top\partial_i X_0^x(t,s).$$

Also, the previous derivation with $\alpha = 1$ and taking the limit $\delta \to 0$ implies that

$$E|\partial_{x_i} h(s, X_0^x(t, s))| \le C(|x| + 1)^m$$

where $C$ doesn't depend on $s$ and $t$.

**Second Space Derivatives:** Then, we show equality (D.2). Previous proof implies that

$$\partial_{x_j} \partial_{x_i} E h(s, X_0(t, s, x)) = \partial_{x_j} E \nabla h(s, X_0^x(t, s))^\top \partial_i X_0^x(t, s).$$

Hence we employ the same strategy to exchange the limit and expectations for the following expression

$$
\begin{aligned}
\lim_{\delta \to 0} E \frac{1}{\delta} & \left[ \nabla h(s, X_0^{x+\delta e_j}(t, s))^\top \partial_i X_0^{x+\delta e_j}(t, s) - \nabla h(s, X_0^x(t, s))^\top \partial_i X_0^x(t, s) \right] \\
= & \lim_{\delta \to 0} E \frac{1}{\delta} \partial_i X_0^x(t, s)^\top (\nabla h(s, X_0^{x+\delta e_j}(t, s)) - \nabla h(s, X_0^x(t, s))) \qquad \text{(D.6)} \\
& + \lim_{\delta \to 0} E \frac{1}{\delta} \nabla h(s, X_0^{x+\delta e_j}(t, s))^\top (\partial_i X_0^{x+\delta e_j}(t, s) - \partial_i X_0^x(t, s))
\end{aligned}
$$

We show U.I. for the two terms in (D.6) separately. For the first term, consider

$$
\begin{aligned}
& E \left| \frac{1}{\delta} \partial_i X_0^x(t, s)^\top (\nabla h(s, X_0^{x+\delta e_j}(t, s)) - \nabla h(s, X_0^x(t, s))) \right|^\alpha \\
& \le \left( E |\partial_i X_0^x(t, s)|^{2\alpha} E \frac{1}{\delta^{2\alpha}} \left| \nabla h(s, X_0^{x+\delta e_j}(t, s)) - \nabla h(s, X_0^x(t, s)) \right|^{2\alpha} \right)^{1/2}
\end{aligned}
$$

By Proposition A.1, the first expectation is bounded uniformly in $s$ and $t$. For the second term, consider

$$
\begin{aligned}
& \frac{1}{\delta^{2\alpha}} \left| \nabla h(s, X_0^{x+\delta e_j}(t, s)) - \nabla h(s, X_0^x(t, s)) \right|^{2\alpha} \\
& = \frac{1}{\delta^{2\alpha}} \left( \sum_{i=1}^d \left| \partial_i h(s, X_0^{x+\delta e_j}(t, s)) - \partial_i h(s, X_0^x(t, s)) \right|^2 \right)^\alpha \\
& \overset{(i)}{\le} \frac{1}{\delta^{2\alpha}} \left( \left| X_0^{x+\delta e_j}(t, s) - X_0^x(t, s) \right|^2 \sum_{i=1}^d \left| \nabla \partial_i h(s, \xi_i X_0^{x+\delta e_j}(t, s) + (1 - \xi_i) X_0^x(t, s)) \right|^2 \right)^\alpha \\
& = \frac{1}{\delta^{2\alpha}} \left| X_0^{x+\delta e_j}(t, s) - X_0^x(t, s) \right|^{2\alpha} \left| H[h](s, \xi_i X_0^{x+\delta e_j}(t, s) + (1 - \xi_i) X_0^x(t, s)) \right|^{2\alpha} \\
& \overset{(ii)}{\le} \frac{c_h^{2\alpha}}{\delta^{2\alpha}} \left| X_0^{x+\delta e_j}(t, s) - X_0^x(t, s) \right|^{2\alpha} (|X_0^{x+\delta e_j}(t, s) - X_0^x(t, s)| + |X_0^x(t, s)| + 1)^{2\alpha m}
\end{aligned}
$$

where $(i)$ follows from the mean value theorem with r.v. $\xi_i \in [0, 1]$, and $(ii)$ applies Assumption 4. Therefore, we have that

$$
\begin{aligned}
& E \frac{1}{\delta^{2\alpha}} \left| \nabla h(s, X_0^{x+\delta e_j}(t, s)) - \nabla h(s, X_0^x(t, s)) \right|^{2\alpha} \\
& \le C E \frac{1}{\delta^{2\alpha}} \left| X_0^{x+\delta e_j}(t, s) - X_0^x(t, s) \right|^{2\alpha(m+1)} \\
& \quad + C \left( E \left[ (|X_0^x(t, s)| + 1)^{4\alpha m} \right] E \frac{1}{\delta^{4\alpha}} \left| X_0^{x+\delta e_j}(t, s) - X_0^x(t, s) \right|^{4\alpha} \right)^{1/2} \\
& \le C \left[ \delta^{2\alpha m} l_{2\alpha(m+1)}^{2\alpha(m+1)} + C(1 + |x|)^{2\alpha m} \right]
\end{aligned}
$$

where the last inequality follows from Proposition A.1. This is uniformly bounded in $\delta$ as $\delta \to 0$, showing U.I. for the first term in (D.6).

For the second term in (D.6), we consider

$$E\left|\frac{1}{\delta}\nabla h(s,X_0^{x+\delta e_j}(t,s))^\top(\partial_i X_0^{x+\delta e_j}(t,s)-\partial_i X_0^x(t,s))\right|^\alpha$$

$$\leq\left(E\left|\nabla h(s,X_0^{x+\delta e_j}(t,s))\right|^{2\alpha}\cdot E\frac{1}{\delta^{2\alpha}}\left|(\partial_i X_0^{x+\delta e_j}(t,s)-\partial_i X_0^x(t,s))\right|^{2\alpha}\right)^{1/2}$$

$$\leq l_{2\alpha}^\alpha c_h^\alpha\left(E(|X_0^{x+\delta e_j}(t,s)|+1)^{2\alpha m}\right)^{1/2}$$

$$\leq C(b_{2\alpha m}^{2\alpha m}(|x+\delta e_j|+1)^{2\alpha m}+1)^{1/2}$$

which is also uniformly bounded in $\delta$ as $\delta\to 0$.

Therefore, exchanging the limits in (D.6), we obtain

$$\partial_{x_j}\partial_{x_i}Eh(s,X_0(t,s,x))=E\partial_i X_0^x(t,s)^\top H[h](s,X_0^x(t,s))\partial_j X_0^x(t,s)+\nabla h(s,X_0^x(t,s))^\top\partial_j\partial_i X_0^x(t,s).$$

Moreover, by setting $\alpha=1$ and taking the limit as $\delta\to 0$ in the preceding derivations, we see that

$$E|\partial_{x_j}\partial_{x_i}h(s,X_0(t,s,x))|\leq C(|x|+1)^m.$$

where the constant $C$ is uniform in $s$ and $t$.

## E    Proof of Theorem 3'

From (A.1), we see that

$$E\int_0^T\nabla_\theta L_0 V_0(t,X_0^x(0,t))dt=E\int_0^T\nabla_\theta\mathcal{L}_0 v_0(t,X_0^x(0,t))dt.$$

Moreover, since $\tau$ is independent of $\mathcal{F}$,

$$E\int_0^T\nabla_\theta L_0 V_0(t,X_0^x(0,t))dt=T\int_0^T E[\nabla_\theta L_0 V_0(\tau,X_0^x(0,\tau))|\tau=t]\frac{1}{T}dt$$
$$=TEE[\nabla_\theta L_0 V_0(\tau,X_0^x(0,\tau))|\tau]$$
$$=ET\nabla_\theta L_0 V_0(\tau,X_0^x(0,\tau))$$

Therefore, by Theorem 1', $ED(x)=\nabla_\theta v_0(0,x)$.

For the variance, we consider

$$E|T\nabla_\theta L_0 V_0(\tau,X_0^x(0,\tau))|^2=\int_0^T E[|T\nabla_\theta L_0 V_0(t,X_0^x(0,t))|^2|\tau=t]\frac{1}{T}dt$$

$$=TE\int_0^T|\nabla_\theta L_0 V_0(t,X_0^x(0,t))|^2 dt$$

$$\leq C\int_0^T E|\nabla_\theta\mu_0|^2|Z(t,X_0^x(0,t))|^2+|\nabla_\theta a_0|^2|H(t,X_0^x(0,t))|^2 dt$$

$$\leq C\frac{1}{T}\int_0^T\left(E|\nabla_\theta\mu_0|^4 E|Z(t,X_0^x(0,t))|^4\right)^{1/2}+\left(E|\nabla_\theta a_0|^4|H(t,X_0^x(0,t))|^4\right)^{1/2}dt$$

$$\leq C\left(\frac{1}{T}\int_0^T E|\nabla_\theta\mu_0|^4 dt\cdot\frac{1}{T}\int_0^T E|Z(t,X_0^x(0,t))|^4 dt\right)^{1/2}$$

$$+C\left(\frac{1}{T}\int_0^T E|\nabla_\theta a_0|^4 dt\cdot\frac{1}{T}\int_0^T E|H(t,X_0^x(0,t))|^4 dt\right)^{1/2}$$

By (B.11) and (B.13) with $\alpha=2$,

$$\frac{1}{T}\int_0^T E|\nabla_\theta\mu_0|^4 dt\leq\frac{1}{T}\sup_{\theta\in\Theta}\frac{1}{|\theta|^4}\int_0^T\kappa_{\theta,0}^4(s)ds\sup_{s\in[0,T]}E(|X_0^x(s)|+1)^4\leq C(|x|+1)^4$$

and similarly
$$\frac{1}{T}\int_0^T E|\nabla_\theta a_0|^4 dt \le C(|x|+1)^8.$$
By definition and Proposition A.1, we have that

$$
\begin{aligned}
E|Z(t, X_0^x(0,t))|^4 &\le CE\int_t^T |\nabla\rho_0|^4|\nabla X_0^x(t,r)|^4 dr + |\nabla g_0|^4|\nabla X_0^x(t,T)|^4 \\
&\le CE\int_t^T (|X_0^x(t,r)|+1)^{4m}|\nabla X_0^x(t,r)|^4 dr + (|X_0^x(t,r)|+1)^{4m}|\nabla X_0^x(t,T)|^4 \\
&\le 2CT \sup_{r\in[0,T]} E(|X_0^x(t,r)|+1)^{4m}|\nabla X_0^x(t,r)|^4 \\
&\le C(|x|+1)^{4m}.
\end{aligned}
$$

Similarly, $E|H(t, X_0^x(0,t))|^4 \le C(|x|+1)^{4m}$. These calculations imply that
$$E|T\nabla_\theta L_0 V_0(\tau, X_0^x(0,\tau))|^2 \le C(|x|+1)^{2m+4}.$$

For the reward rate and terminal reward terms, we recall Assumption 10 with the additional Assumption that $\alpha > 2$. Note that since $\alpha > 2$, for
$$
\begin{aligned}
|\rho_\theta(t,x) - \rho_0(t,x)|^2 &= |\rho_\theta(t,x) - \rho_0(t,x)|^{\alpha \cdot \frac{2}{\alpha}} \\
&\le \kappa_{\theta,0}^\alpha(s)^{2/\alpha}(|x|+1)^\alpha.
\end{aligned}
$$

So, we have that

$$
\begin{aligned}
E\int_0^T |\nabla_\theta\rho_0|^2 dt &\overset{(i)}{\le} \lim_{\theta\to 0} E\int_0^T \frac{1}{|\theta|^2}|\rho_\theta - \rho_0|^2 dt \\
&\le \sup_{\theta\in\Theta}\int_0^T E\frac{1}{|\theta|^2}|\rho_\theta - \rho_0|^2 dt \\
&\le \sup_{\theta\in\Theta}\int_0^T \frac{1}{|\theta|^2}\kappa_{\theta,0}^\alpha(s)^{2/\alpha} dt \sup_{t\in[0,T]} E(|X_0^x(0,t)|+1)^{2m} \\
&\overset{(ii)}{\le} \left(\sup_{\theta\in\Theta}\int_0^T \frac{1}{|\theta|^2}\kappa_{\theta,0}^\alpha(s) dt\right)^{2/\alpha} C(|x|+1)^{2m} \\
&\le C(|x|+1)^{2m}
\end{aligned}
$$

where $(i)$ uses $\alpha > 0$ so that the integrand is U.I. in $\theta \in \Theta$ (see (B.11) for a similar proof), and $(ii)$ uses Jensen's inequality with $2/\alpha < 1$. The same holds for the terminal reward term, with $\kappa_{\theta,\theta'}^\alpha = \ell^\alpha|\theta - \theta'|^\alpha$ integrable.

Therefore, we conclude that $\text{Var}(|D(x)|) \le E|D(x)|^2 \le C(|x|+1)^{2m+4}$, where $C$ can be dependent on other parameters but not $x$.

## F   Supplementary Materials for Section 4

### F.1   Calculating the Estimators

**The Generator Gradient Estimator:**   We compute

$$
\begin{aligned}
\partial_i v(t,x) &= E\int_t^T \partial_{x_i}\rho_\theta(s, X_\theta^x(t,s))ds + \partial_{x_i}g(X_\theta^x(t,T)) \\
&= E\int_t^T u_\theta(s, X_\theta^x(t,s))^\top(R + R^\top)\nabla u_\theta(s, X_\theta^x(t,s))\partial_i X_\theta^x(t,s)ds \\
&\quad + \int_t^T X_\theta^x(t,s)^\top(Q + Q^\top)\partial_i X_\theta^x(t,s)ds + X_\theta^x(t,T)^\top(Q_T + Q_T^\top)\partial_i X_\theta^x(t,T).
\end{aligned}
$$

Here $\partial_i X_\theta^x(t, s)$ is a column vector. So, replacing it by the Jacobian will yield a row vector. Following the definition of $Z$ in (2.6), we define

$$Z(t,x)^\top := \int_t^T u_\theta(s, X_\theta^x(t,s))(R + R^\top)\nabla u_\theta(s, X_\theta^x(t,s))\nabla X_\theta^x(t,s)ds$$
$$+ \int_t^T X_\theta^x(t,s)^\top (Q + Q^\top)\nabla X_\theta^x(t,s)ds + X_\theta^x(t,T)^\top(Q_T + Q_T^\top)\nabla X_\theta^x(t,T). \tag{F.1}$$

Here, the derivative process $\nabla X_\theta^x$ satisfies the following ODE with random coefficients:

$$\partial_i X_\theta^x(t,s) = e_i + \int_t^s (A + B\nabla u_\theta(r, X_\theta^x(t,r)))\partial_i X_\theta^x(t,r)dr;$$

or in matrix form:

$$\nabla X_\theta^x(t,s) = I + \int_t^s (A + B\nabla u_\theta(r, X_\theta^x(t,r)))\nabla X_\theta^x(t,r)dr.$$

Therefore, in this setting, our generator gradient estimator in (2.9) is

$$D_i(x) = T\partial_{\theta_i} u_\theta(\tau, X_\theta^x(\tau))^\top B^\top Z(\tau, X_\theta^x(\tau)) + Tu_\theta(\tau, X_\theta^x(\tau))^\top(R+R^\top)\partial_{\theta_i}u_\theta(\tau, X_\theta^x(\tau)) \tag{F.2}$$

where $Z$ is given by (F.1). As explained in (2.9), we also randomize the integral corresponding to the gradient of the reward rate $\nabla_\theta \rho_0$.

**The Pathwise Differentiation Estimator:** From (1.3), we construct the following IPA estimator that randomizes the time integral

$$\widetilde{D}_i(x) = Tu_\theta(\tau, X_\theta^x(\tau))(R + R^\top)\nabla u_\theta(\tau, X_\theta^x(\tau))\partial_{\theta_i}X_\theta^x(\tau) + TX_\theta^x(\tau)^\top(Q + Q^\top)\partial_{\theta_i}X_\theta^x(\tau)$$
$$+ Tu_\theta(\tau, X_\theta^x(\tau))^\top(R + R^\top)\partial_{\theta_i}u_\theta(\tau, X_\theta^x(\tau)) + X_\theta^x(T)^\top(Q_T + Q_T^\top)\partial_{\theta_i}X_\theta^x(T).$$

Here, the pathwise derivatives $\partial_{\theta_i}X_\theta^x(t)$ is the solution the following ODE with random coefficient:

$$\partial_{\theta_i}X_\theta^x(t) = \int_0^t (A + B\nabla u_\theta(s, X_\theta^x(s)))\partial_{\theta_i}X_\theta^x(s) + B\partial_{\theta_i}u_\theta(s, X_\theta^x(s))ds. \tag{F.3}$$

### F.2 Numerical Experimentation Details

We conducted the computation time and variance comparison for both estimators using PyTorch. The computation time data was generated on a system equipped with a PCIE version of Nvidia Tesla V100 GPU, featuring 32GB of VRAM. Additionally, the system includes a 2-core CPU and 16GB of RAM, which are used to format and store data. The primary computational tasks are handled by the GPU.

The data for Table 2 is produced as follows. For each $n$, we produce 400 i.i.d. GG and PD estimators $\left\{D^{(j)}(x_0), \widetilde{D}^{(j)}(x_0) \in \mathbb{R}^n : j = 1, \dots, 400\right\}$. Let

$$\hat{\sigma}_{\text{GG},i} := \frac{1}{20}\sum_{j=1}^{400}\left(D_i^{(j)}(x_0) - \frac{1}{400}\sum_{j=1}^{400}D_i^{(j)}(x_0)\right)^2,$$

$$\hat{\sigma}_{\text{PD},i} := \frac{1}{20}\sum_{j=1}^{400}\left(\widetilde{D}_i^{(j)}(x_0) - \frac{1}{400}\sum_{j=1}^{400}\widetilde{D}_i^{(j)}(x_0)\right)^2.$$

The "Avg SE of GG" and "Avg SE of PD" entries record

$$\frac{1}{n}\sum_{i=1}^{n}\hat{\sigma}_{\text{GG},i} \quad \text{and} \quad \frac{1}{n}\sum_{i=1}^{n}\hat{\sigma}_{\text{GG},i}, \tag{F.4}$$

respectively. The "Avg SE ratios" compute

$$\frac{1}{n}\sum_{k=1}^{n}\frac{\hat{\sigma}_{\text{GG},i}}{\hat{\sigma}_{\text{PD},i}}. \tag{F.5}$$

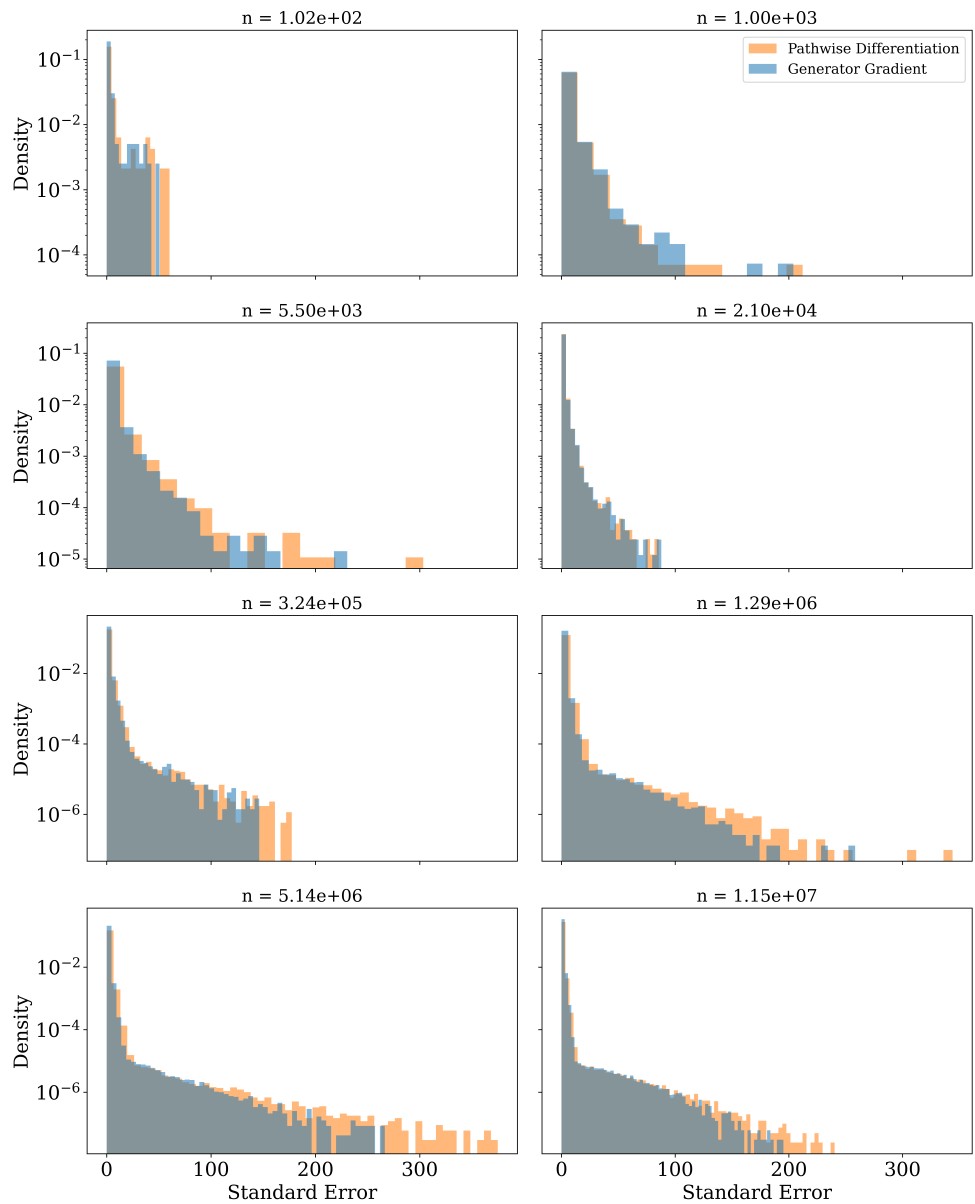

Figure 2: Histograms comparison of the distribution formed by the standard errors of coordinates of the estimators. These plots use the same data as that produces Table 2.

The numerical values used for the matrices, initial conditions, and network initializations for the SDE models can be found in the supplied code.

We further analyze variance by plotting histograms of the distribution formed by the standard errors of the coordinates of the estimators, as shown in Figure 2. The standard error distribution of the pathwise differentiation method exhibits a heavier tail compared to our proposed generator gradient estimator. This aligns with the superior variance performance of our estimator demonstrated in Table 2. Figure 2 also provides insights into the confidence intervals in Figure 1b, which are barely visible due to high confidence levels. In particular, the generator gradient estimator has tighter confidence intervals in Figure 1b.

# G    Experiments on SDEs with Non-Differentiable Parameters

## G.1    CIR Model

In this section, we use the Cox-Ingersoll-Ross (CIR) diffusion as an example to test the validity of the proposed generator gradient estimator when the differentiability assumptions of the coefficients are violated. Specifically, consider the one-dimensional process:

$$X_\theta^x(t, s) = x + \int_t^s (\theta - X_\theta^x(t, r))dr + \int_t^s \sqrt{X_\theta^x(t, r)}dB(r) \tag{G.1}$$

for $t, s \in [0, 2]$, where $x, \theta > 0$. Note that the volatility $\sigma(t, x) = \sqrt{x}$ is not differentiable at 0, though it is $C^\infty$ for $x > 0$. A unique strong solution to (G.1) always exists. Moreover, if $\theta \geq 1/2$, then $X_\theta(t) > 0$ for all $t \in [0, 2]$ almost surely.

We consider the following value function:

$$v_\theta(t, x) := E\left[\int_t^T X_\theta^x(t, s)ds\right]. \tag{G.2}$$

We aim to estimate the gradient $\nabla_\theta v_\theta(0, x)$ evaluated at $x = 0.1$ for multiple values of $\theta$.

Since the pathwise differentiation estimator also suffers from non-differentiability issues, we validate the generator gradient (GG) estimator by comparing it with the finite difference (FD) estimator. Specifically, the FD estimator computes

$$\frac{1}{h}\Delta_h(\theta) := \frac{1}{h}\left[v_{\theta+\frac{h}{2}}(0, x) - v_{\theta-\frac{h}{2}}(0, x)\right] \tag{G.3}$$

using Monte Carlo simulation of the SDE (G.1) and the value function. In this context, the FD estimator should consistently estimate the gradient evaluated at any $\theta > 0$. The GG estimator is produced from (2.9). We use the Euler scheme to simulate the SDEs and the derivative processes. To avoid numerical issues when the Euler discretization of the CIR process crosses 0, we take the absolute value of the discretized process at each time step.

Table 3: Statistics for $10^6$-sample averaged GG and FD estimator. For the FD estimator, we choose $h = 0.05$ in (G.3), resulting in a bias of $O(h^2)$.

| Value of $\theta$ | 4 | 2 | 0.55 |
|---|---|---|---|
| GG $\pm$ 95% CI | $1.135 \pm 0.0011$ | $1.135 \pm 0.0012$ | $1.134 \pm 0.0019$ |
| FD $\pm$ 95% CI | $1.095 \pm 0.064$ | $1.097 \pm 0.046$ | $1.116 \pm 0.026$ |

| Value of $\theta$ | 0.45 | 0.2 | 0.1 |
|---|---|---|---|
| GG $\pm$ 95% CI | $1.134 \pm 0.0023$ | $1.179 \pm 0.112$ | $1.492 \pm 9.759$ |
| FD $\pm$ 95% CI | $1.112 \pm 0.024$ | $1.020 \pm 0.018$ | $0.895 \pm 0.015$ |

Table 3 summarizes the estimated value and confidence interval for both the GG and FD estimators. We note that a bias of $O(h^2)$ is present in the FD case. When $\theta \geq 1/2$, we observe that even though Assumptions 1 and 3 are violated, the GG estimator produces results consistent with the FD estimator. This suggests the validity of the GG estimator even when Assumptions 1 and 3 don't hold. This consistency occurs because, in this case, the derivative processes are still well-defined up to the first time $X_\theta^x(t)$ hits 0, which does not happen when $\theta > 1/2$.

However, when $\theta < 1/2$ (cases highlighted in blue in Table 3), the sample paths of the SDE (G.1) can reach 0. Although the statistics in Table 3 appear consistent, we observe a significant increase in the variance of the GG estimator as $\theta$ decreases. This increase may indicate that the GG estimator is not consistently estimating the gradient in these cases.

## G.2 SDE with ReLU Drift

In this section, we use the following SDE with ReLU drift as an example to test the validity of the proposed generator gradient estimator:

$$X_\theta^x(t, s) = x + \int_t^s (\text{ReLU}(\theta X_\theta^x(t, r)) + 1)dr + \int_t^s dB(r) \tag{G.4}$$

for $t, s \in [0, 2]$, where $\theta > 0$ and we choose $x = -0.1$. Note that the $+1$ in the drift makes it always positive. So, starting from $-0.1$, the process should cross 0 (where the drift is non-differentiable) before time 2 with high probability.

With $v_\theta$ defined in (G.2), we aim to estimate the gradient $\nabla_\theta v_\theta(0, x)$ evaluated at $x = 0.1$ for multiple values of $\theta$ using the GG and FD (defined in (G.3)) estimators.

Table 4: Statistics for $10^6$-sample averaged GG and FD estimators. For the FD estimator, we choose $h = 0.05$ in (G.3), resulting in a bias of $O(h^2)$.

| Value of $\theta$ | 2 | 1 | 0.5 |
|---|---|---|---|
| GG $\pm$ 95% CI | $14.91 \pm 0.031$ | $4.087 \pm 0.008$ | $2.300 \pm 0.005$ |
| FD $\pm$ 95% CI | $14.78 \pm 0.570$ | $4.131 \pm 0.192$ | $2.394 \pm 0.127$ |

Table 4 summarizes the estimated values and confidence intervals for both the GG and FD estimators. Note that a bias of $O(h^2)$ is present in the FD case. Despite violations of Assumptions 1 and 3, the GG estimator still produces consistent results compared to the FD estimator. We note that in this context, it should be possible to establish the existence and integrability of the derivative processes for any value of $\theta$.

