# OpenReview forum: "An Efficient High-dimensional Gradient Estimator for Stochastic Differential Equations"
_NeurIPS.cc/2024/Conference — NeurIPS 2024 poster_

### Official Review · Reviewer_5TVX · 2024-06-14

**Soundness:** 4
**Presentation:** 2
**Contribution:** 3
**Rating:** 4
**Confidence:** 4

**Summary:**

The article studies classical stochastic control in the continuous d-dimensional SDE setting (with and without jumps). The authors suggest to compute the gradients of the value function (expected total reward in the non-discounted finite time regime), which can be applied (in a model-based setting) to policy gradient type algorithms popular in RL. While in a straight-forward fashion computing the derivatives of the expectations would lead to the simulation of n (dimension of approximation space for the control) expectations that require the simulation of one SDE each, the authors propose a clever interchange of derivatives that reduces the number of SDEs to a polynomial in d. Very careful proofs are given, the findings are applied to simple linear SDE example with quadratic cost function.

**Strengths:**

The article introduces a creative switching trick (switching order of derivatives and derivatives with expectations) that allows to reduce the simulation dimension compared to a straight-forward estimator.

The rest of the paper consists of a rigorous verification of interchanging limits (derivatives) and expectations. The proofs are careful (mainly based on differentiability properties of SDE flows), I spotted only few typos.

**Weaknesses:**

I do not agree with the comparison to RL (=sample based stochastic control) in Section 1.1. The entire point of RL is to provide model-free estimators, in particular, the policy gradient theorem cancels out the model transitions. Here, the situation is different. The model ($\mu$ and $\sigma$) is used to write down the SDEs (2.7) required to simulate the estimator. This is not a problem but shifts the article more towards the stochastic control community.

I cannot see a clear interest in the ML/AI community in the questions raised, the control setting of SDEs (jump SDEs) is natural in the probability/math finance community but not at all in ML/AI. Even if I try I struggle to come up with examples relevant for the ML/AI community (the authors neither).

The article is written for a purely mathematical audience with research level skills in stochastic process theory, the article could appear almost unchanged in journals such as SPA or SIAM Journal on Control and Optimzation. I think NeurIPS is not the right place for publication. I am willing to increase my scores if a more relevant example than the quadratic problem could be provided that shows value of the switching trick to the ML/AI community.

The appendix is chaotic. It is tricky to follow the proofs because parts of the proofs are deferred to other sections of the appendix. Given that the proof only justifies interchange of limits and expectations it looks much harder than it is.

**Questions:**

1. You will typically have to solve the three SDES using numerical schemes. Can you provide additional assumptions on the model (the coefficients) that, combined with SDE numeric results (let's say in the Brownian case only), provides a result for the estimation error of the gradient?

2. The additional derivatives on the model coefficients make me doubt the SDE scheme (2.7) works well if the coefficients are not so nice. Many control problems in applications are Heston type models, which are already problematic for numerics with the square-root variance coefficient. I know they do not satisfy the assumptions imposed, did you still try your simulation scheme?

3. I do not understand why you included the jumps without an application in mind. This is normal for a maths journal, but a bit strange in ML/AI. Could you please provide an example?

4. A few minor points:
(i) The generator of a Markov process is the action plus the domain. Could you please be a bit more careful when you rework the article?
(ii) You might want to mention that uniformly bounded derivatives imply linear growth which is the standard property required for global existence.
(iii) I am probably missing where you discuss properties on the model that imply Assumption 2. If not I find assumptions on the derivatives of the mean rewards disturbing that I do not know how to check.
(iv) Typo in Assumption 1, 2.
(v) At least for the larger expectations it would help to be more consistent with the use of brackets.
(vi) Could you please put references to the quadratic loss examples? Since this is the only application it would be good to see where and how it is used.
(vii) the use of colons is not very helpful, for instance in (2.6) it is a bit complicated to understand what object is defined (I guess you want to define Z).
(viii) The chain rule is missing in (D.3)

**Limitations:**

The authors did not discuss the limitations or the practical problems of the results.

---

> ### Author Rebuttal · Authors · 2024-08-07
>
> We thank the reviewer for the insightful comments. We hope that the following response will help to address some of the concerns.
>
> 1. **Applicability and interest to the ML community:** Recently, there has been significant interest in approximating continuous-time optimal control using neural networks and gradient methods, with applications in finance [1], scheduling of queuing systems [2], and stabilizing stochastic systems [3]. Our method offers a new scalable approach to approximate the optimal policies of these complex stochastic control problems. Additionally, our methodology can be applied to many ML-in-science contexts, including PDE-constrained optimization [4] and neural SDEs [5,6,7].
>
>
> 2. **Stochastic control and RL:** We note that our literature review in Section 1.1 aims to acknowledge and compare previous research on Monte Carlo gradient estimation and its applications. We consider both policy gradient methods in RL and gradient methods in stochastic control to be important examples. However, we do not intend to claim that they are equivalent. We will clarify this by improving the writing in this section.
>
>
> 3. **Applications of jump diffusion models:** Regarding the reviewer's concern about the applicability of diffusions with jumps, we note that many models in the finance literature require the inclusion of jumps to accurately capture market behavior [8] (cited in the paper). Therefore, ML models for financial applications would benefit from the versatility of including jumps. Moreover, continuous-time Markov chain (CTMC) models are widely used in chemistry, biology, and physics to model the stochastic behavior of natural systems (see [7] and references therein, also cited in the paper). CTMCs are also a valuable tool for analyzing the behavior of discrete event systems. The jump-diffusion model considered in this paper captures a large class of CTMCs, underscoring the importance of including jump behavior in the SDEs we study.
>
>
> 4. **Assumptions on the smoothness of SDE coefficients:** We address the concerns regarding numerical error analysis and the applicability of our methodology when the SDE coefficients are not smooth in the "Author Rebuttal" section. In particular, we have tested our estimator for the CIR process with a square-root volatility, yielding results that are consistent with our expectations. We believe that the generator gradient estimator remains unbiased with finite variance for a wide range of non-smooth SDE models, and we support this claim with numerical experiments. We hope these additions will alleviate some of the concerns.
>
>
> 5. **Typos and the organization of the Appendices:**  We thank the reviewer for their careful review and for pointing out typos and presentation issues within this paper. We will address these issues in the revision. Regarding the organization of the appendices, we aim to first present an overview of the important supporting results (presented as lemmas and propositions) and discuss why proving these results will imply the main theorems. We defer the detailed proofs of these supporting results to the appendices for interested readers. These proofs are typically rigorous checks for the uniform integrability of the pre-limit expectations and are lengthy due to the presence of many terms. We note that in order to get the correct polynomial growth order of the variance in Theorem 3, a careful analysis of the growth power of the parameters and rewards is required.
>
> 6. **Establishing Assumption 2:** Regarding the difficulty of checking Assumption 2, it can be established using a stochastic flow argument as in [Kunita 2019, Chapter 4] (cited in our paper) if the rewards are sufficiently smooth and the derivative processes are integrable. This can also be established using PDE arguments (see [Evans 2022], also cited in our paper). However, due to space limitations, we didn't include a full discussion of how this is done and instead presented it as an assumption. This is discussed in lines 196-201 after Assumption 2.
>
> Finally, we have decided to submit this manuscript for publication in the proceedings of NeurIPS to share our research output with a broader community. We believe that our methodology will find exciting applications beyond our domain of expertise. This aligns with the spirit of NeurIPS.
>
> We believe that these additional discussions and clarifications enhance the quality and clarity of the paper. If our response addresses your concerns, we kindly request that you consider raising your score.
>
> [1] Fan, Lei, and Justin Sirignano. Machine Learning Methods for Pricing Financial Derivatives. arXiv preprint arXiv:2406.00459 (2024).
>
> [2] Ata, Baris, and Ebru Kasikaralar. Dynamic Scheduling of a Multiclass Queue in the Halfin-Whitt Regime: A Computational Approach for High-Dimensional Problems. Available at SSRN 4649551 (2023).
>
> [3] Zhang, Jingdong, Qunxi Zhu, and Wei Lin. Neural stochastic control. Advances in neural information processing systems 35 (2022): 9098-9110.
>
> [4] Sirignano, Justin, Jonathan MacArt, and Konstantinos Spiliopoulos. PDE-constrained models with neural network terms: Optimization and global convergence. Journal of Computational Physics 481 (2023): 112016.
>
> [5] Tzen, Belinda, and Maxim Raginsky. Neural stochastic differential equations: Deep latent gaussian models in the diffusion limit. arXiv preprint arXiv:1905.09883 (2019).
>
> [6] Kidger, Patrick. On neural differential equations. arXiv preprint arXiv:2202.02435 (2022).
>
> [7] Jia, Junteng, and Austin R. Benson. Neural jump stochastic differential equations. Advances in Neural Information Processing Systems 32 (2019).
>
> [8] Merton, Robert C. Option pricing when underlying stock returns are discontinuous. Journal of financial economics 3.1-2 (1976): 125-144.

---

> > ### Comment · Reviewer_5TVX · 2024-08-08
> > **Thanks for answering my questions!**
> >
> > Thanks for answering my questions!
> >
> > I see in the other reviews good interest in PDE constraint optimization. I cannot judge this direction and it seems this is already largely reflected in the scores of other reviewers. From the point of view of RL and also Math Finance I am still not convinced. ML has not proved much relevance in finance and I do not believe it will in the future. There are plenty of numerical approaches towards affine processes (CIR was really tortured with all possible techniques). Without seeing a proper comparison I cannot believe that the (super inefficient!) policy gradient approach can add any value in this model-based setting. It could in the model-free setting which is not possible with the present approach.
> >
> > I still do not understand why there is no proof for Assumption 2 (in the appendix). If you can prove it, prove it. And identify clearly a set of necessary conditions. Also I do not see a reason not to include a numerical analysis, this would make the paper much stronger. The contribution of only the nice (!) main idea and justifying some changes of limits that everyone in the ML community would believe anyways is a bit too small for NeurIPS, given a practical need is not completely clear. I will keep my score.

---

> > > ### Author Response · Authors · 2024-08-10
> > > **Comments to the reviewer's feedback**
> > >
> > > We thank the reviewer for your quick feedback. We have the following response.
> > >
> > > **Application of Machine learning in finance:**
> > >
> > > We respectfully disagree with the comment that machine learning methods have "not proved much relevance in finance." In fact, the use of artificial neural networks for financial SDE modeling and optimization dates back to the early 1990s. Survey paper [9] summarizes **more than 150 papers** that use neural network-parameterized SDEs to model prices in option pricing and hedging. Additionally, recent papers surveyed in [9] actively explore the use of deep network architectures. This underscores the relevance of our proposed methodology in cutting-edge finance applications. Furthermore, based on the private consulting activities of some of the authors, neural networks and other machine learning techniques are very much alive and of significant interest in financial applications.
> > >
> > > Given these academic interests and industry experiences, we believe it is hard to argue otherwise. *We respectfully ask why you think that ML has not proven to be relevant in finance.*
> > >
> > > **Affine processes:**
> > >
> > > We do not intend to use our method to optimize simple affine processes. Our methodology is designed for estimating sensitivity or optimizing values driven by high-dimensionally parameterized jump-diffusions. Our method excels in modern ML settings where deep neural networks are used in the SDE parameters.
> > >
> > > **Application in sciences and engineering:**
> > >
> > > Other than financial applications, our methodology is well-motivated and an important contribution considering its application in science and engineering. For example, neural SDEs aim to learn the behavior of natural (physical) random processes with parameterized SDEs. Highly cited papers in this field include [7] and [5] (with and without jump). Neural SDEs typically optimize $\min_\theta E[g(X_\theta^x(T))]$ subject to an SDE
> > > $$ dX_\theta^x(t) = \mu_\theta(t, X_\theta^x(t)) dt + \sigma_\theta(t, X_\theta^x(t))  dB(t) + \chi_\theta(t,X_\theta^x(t))dN(t)$$
> > > where $\mu_\theta,\sigma_\theta,\chi_\theta$ are neural network parameterized and $N(t)$ is a Poisson process.
> > >
> > > Gradient methods for approximating stochastic optimal control with a parameterized policy class [3] are also important applications.
> > >
> > > We remark that many neural network-parameterized SDEs are not widely used primarily because of the lack of scalable gradient estimators. Our contribution directly addresses this issue.
> > >
> > > Given these numerous applications across multiple disciplines, we find it **hard to believe** that NeurIPS, where practitioners and theorists, scientists and engineers exchange ideas and advance machine learning theory and applications, is not a suitable venue to publish this work.
> > >
> > > **Rigorous validation of limit interchange:**
> > >
> > > We respectfully disagree with the claim that the interchange of limits is what "everyone in the ML community would believe anyways." Wrong intuition regarding the interchange of limits can lead to serious errors. A researcher might explore the following gradient estimation idea: From semigroup theory, write $E_x[g(X_{\theta}(t))] = (e^{tL_{\theta}}g)(x)$. Formal differentiation (taking the limit quotient) yields: $$\partial_\theta (e^{tL_{\theta}}g)(x) \stackrel{?}{=} t e^{tL_{\theta}} \partial_\theta L_{\theta} g(x) = t E_x[\partial_{\theta}L_{\theta} g(X(t))].$$ Although this can work for Markov jump processes, it is incorrect, unfortunately, for diffusions. The correct expression is precisely the representation in our Theorem 1. The reason readers find the ideas in Section 2 natural is that we have invested research effort into identifying a simple and intuitive justification, all backed by rigorous proof.
> > >
> > > Nevertheless, given this helpful discussion with the reviewer, we think including this incorrect reasoning in the paper could help to illustrate the need for caution when working with generators of SDEs.
> > >
> > > **Assumption 2:**
> > >
> > > We have considered including sufficient conditions on the model primitives to imply Assumption 2 but decided against it for the following reason: The primary focus of this paper is to rigorously justify the **validity** of the generator gradient estimator. Proving Assumption 2 using sufficient conditions could detract from this goal.
> > >
> > > The differentiability and moment bounds for the value are well-established in the literature. Also, justifying Assumption 2 from model primitives would introduce another set of assumptions typical in jump-diffusion analysis but potentially confusing for users, as we have explained in the previous response.
> > >
> > > If you still have concerns about establishing Assumption 2, we can provide a proof in the Appendix with smooth and bounded rewards (so that there will be no interference with the variance's growth power).
> > >
> > > [9] Ruf, Johannes, and Weiguan Wang. Neural networks for option pricing and hedging: a literature review. Journal of Computational Finance.

---

> > > > ### Comment · Reviewer_5TVX · 2024-08-12
> > > > **Thanks for your answers!**
> > > >
> > > > Thanks for your extensive answers to my concerns!

---

### Official Review · Reviewer_Eiq9 · 2024-07-05

**Soundness:** 4
**Presentation:** 4
**Contribution:** 4
**Rating:** 7
**Confidence:** 4

**Summary:**

The paper proposes a novel algorithm for the gradient of machine learning objectives that are based on SDE paths, w.r.t. parameters. The method is scalable for big parameters, as the computational complexity of the estimation is not related to the number of parameters. Empirical results show the scalability advantage over classical pathwise differentiation.

**Strengths:**

PDE constrained optimization has remained a very difficult question due to the high cost of simulating PDEs, let alone taking derivatives. Therefore, despite its many good theoretical promises, it has not been deployed. This paper attacks the question by reducing the computational cost with a monte-carlo estimation of the derivative, largely advances the promise of PDE constrained optimization.
The writing is also very good, and the intuitions behind the methods are well illustrated.

**Weaknesses:**

1. Despite the fact that the computational complexity is irrelevant to the number of parameters, I suspect that as the number grows, the method would require more samples for an accurate monte-carlo estimate of the gradients.
Is it possible to either:
1. Provide a theoretical analysis on the convergence rate of the estimation w.r.t. the number of parameters.
2. Provide an empirical analysis on the convergence rate of the estimation w.r.t. the number of parameters.

**Questions:**

See weakness. Also, applying the Feynman-Kac twice requires the objective to be very smooth, can you provide some insights on the applications where this holds true?

**Limitations:**

As written in questions, the smoothness requirements induced by Feynman Kac etc. will limit the use cases.
As written in weaknesses, it is still unclear the convergence w.r.t. high number of parameters.

---

> ### Author Rebuttal · Authors · 2024-08-07
>
> We thank the reviewer for the comments and questions. We hope that the following response can address some of your concerns:
>
> 1. **Limitations induced by the assumptions on the smoothness:**
>    In the "Author Rebuttal" section, we address concerns regarding the clear presentation of the limitations due to requiring additional smoothness in the SDE parameters, as well as the proof techniques we can adopt to circumvent the theoretical challenges when smoothness in the SDE parameters is lacking. This is also confirmed by our additional empirical demonstration using the CIR process and SDE with ReLU drift. These new experimental results suggest that our generator gradient estimator remains consistent with finite variance even when the SDE parameters are not smooth, indicating a wide applicability of our proposed methodology.
>
>    Regarding the application of the Feynman-Kac formula, we use a probabilistic proof based on martingale methods. This approach requires applying the Feynman-Kac formula once to obtain the martingales in Lemma 1, taking the limit of the difference quotient, and proving uniform integrability to get the $\theta$ gradient. The introduction that uses PDE techniques is only for intuitive understanding. We note that probabilistic tools can sometimes yield weaker smoothness requirements than functional analytic methods.
>
>     That said, our approach does require that the value function is a classical solution (with possible relaxation that the second derivative is continuous almost everywhere) to apply Itô's lemma and the Feynman-Kac formula (see Assumption 2). Nevertheless, Assumption 2 can be established using a stochastic flow argument as in [Kunita 2019, Chapter 4] (cited in our paper) if the rewards are sufficiently smooth and the derivative processes are integrable. It can also be established using PDE arguments (see [Evans 2022], also cited in our paper). Due to space limitations, we did not include a full discussion of how this is done and instead presented it as an assumption. This is discussed in lines 196-201 after Assumption 2.
>
>
> 2. **Convergence properties as the number of parameters increases:** The variance behavior of our algorithm, when the number of parameters becomes very large, could be a concern. Therefore, we included Figure 2 in Appendix F.2, showing histograms of the standard errors of each of the gradient coordinates under a wide range of $n \in [10^2, 10^8]$. We see that the tail of the standard errors grows from about 25 when $n \approx 100$ to about 200 when $n \approx 10^7$. We believe that this is not a prohibitive increase in variance, considering that the number of parameters is many orders of magnitude larger. This, coupled with the sketched error analysis in the "Author Rebuttal" section, suggests that the increase in variance is manageable. Additionally, the generator gradient estimator has a lighter tail compared to the pathwise differentiation estimator, suggesting a uniformly better variance performance in this setting.
>
>     However, we should note that the standard error distribution depends on the $\theta$ where we evaluate the neural network. Here, we use the default initialization provided by PyTorch.
>
>     Even though the generator gradient estimator doesn't need to simulate an SDE of dimension linear in $n$, we still need to evaluate the neural network and compute $\nabla_\theta \mu_\theta$ and $\nabla_\theta \sigma_\theta$, which could be costly if $n$ is large. However, upon investigation, we find that our proposed estimator $D(x)$ evaluates $\nabla_\theta \mu_\theta$ and $\nabla_\theta \sigma_\theta$ only once. In contrast, using pathwise differentiation requires evaluating $\nabla_\theta \mu_\theta$ and $\nabla_\theta \sigma_\theta$ at each time step to simulate $\nabla_\theta X_\theta^x(t)$. We believe this is why our estimator has a very stable computation time even when $n \approx 10^7$.
>
> 3. **Error bounds:** As we have explained in the "Author Rebuttal" section, the typical worst-case upper bounds produced by analyzing the numerical SDE schemes, applied to the derivative processes, while reassuring the validity of the proposed estimator, cannot explicitly demonstrate the variance and bias dependence on the dimensions. Moreover, we believe that the error's dependence on these dimensions is instance-specific, depending on the SDE and reward functions. These dependencies are technically challenging to explicitly reflect in an error bound. We hope that the previous empirical validation of the variance behavior can alleviate your concern.
>
> We believe that these additional discussions and clarifications enhance the quality and clarity of the paper. Additionally, we are confident that our methodology will attract interest from various research communities, and sharing our work at NeurIPS will help achieve this goal. If our response addresses your concerns, we kindly request that you consider raising your score.

---

> > ### Comment · Reviewer_Eiq9 · 2024-08-09
> >
> > Thanks for your response. I will keep my score as is.

---

### Official Review · Reviewer_rGnk · 2024-07-13

**Soundness:** 4
**Presentation:** 4
**Contribution:** 3
**Rating:** 7
**Confidence:** 4

**Summary:**

This paper formulates an efficient, unbiarsed, and finite variance gradient estimator for an objective function that looks like the stochastic optimal control cost function (it is ubiquitous across various applications). The problem concerns overparametrized SDEs (the paramete dimension n r is of a much higher dimension than the state space d).  The algorithm runs at a complexity invariant to the large n. The authors use the Feynman-Kac PDE for v_{\theta} and transform it to an equivalent PDE to solve. They then employ a pathwise differentiation estimator to estimate the 2nd order and 1st order derivatives in the PDE while exploiting certain properties of the Hessian.

**Strengths:**

1. The applications of gradient estimators and the motivation behind this problem is well stated and supported.
2. The methodology is clear, especially when finding probablistic representations of the gradient estimator.

**Weaknesses:**

1. The authors discuss the limitations in the checklist but it would have been more clear to have a limitations section clearly outlined as claimed in the checklist.
2. The assumptions may be restrictive as mentioned.
3. The work could be improved if there was a numerical simulation. The example of LQR is an illustrative example and it may be of didactic use and potentially relatively easy to implement a scenario when the parametrization is of a much higher dimension.
4. The spatial derivatives \partial v_0 are still difficult to compute, but still better than \partial v_{\theta}. What are some error bounds you can get by using the later derived probabilistic representations that can use Monte Carlo

**Questions:**

1. Are there any error bounds from using the probabilistic representations for the spatial derivatives (see #4 in Weaknesses)

Happy to revise score if necessary.

**Limitations:**

Yes they have mentioned some, rest may be in the above Weakness section unless there are some misunderstandings.

---

> ### Author Rebuttal · Authors · 2024-08-07
>
> We thank the reviewer for the comments and questions. We organize our responses as follows
>
> 1. **Limitations induced by the assumptions on the smoothness:** Thanks for the suggestion of including a separate discussion of limitations to avoid misunderstanding. In the "Author Rebuttal" section, we address concerns regarding the clear presentation of the weaknesses induced by requiring additional smoothness in the SDE parameters, as well as the proof techniques we can adopt to circumvent the theoretical challenges when smoothness in the SDE parameters is lacking. This is also confirmed by our additional empirical demonstration using the CIR process and SDE with ReLU drift. These new experimental results suggest that our generator gradient estimator remains consistent with finite variance even when the SDE parameters are not smooth, indicating a wide applicability of our proposed methodology.
>
> 2. **Error bounds:** As we have explained in the "Author Rebuttal" section, the typical worst-case upper bounds produced by analyzing the numerical SDE schemes, applied to the derivative processes, while reassuring the validity of the proposed estimator, cannot explicitly demonstrate the variance and bias dependence on the dimensions. Moreover, we believe that the error's dependence on these dimensions is instance-specific, depending on the SDE and reward functions. These dependencies are technically challenging to explicitly reflect in an error bound.
>
> 3. **High-dimensional parameterization:** For the LQG example, we have tested it in an extremely overparameterized setting where $n$ the dimension of $\theta$ is more than $10^7$. In this setting, our estimator outperforms the pathwise differentiation estimator in computation time as well as variance. We can also provide a demonstration of running gradient descent using our estimator, plotting the terminal performance of the control policy. If this could resolve your concern, please let us know.
>
> We believe that these additional discussions and clarifications enhance the quality and clarity of the paper. Additionally, we are confident that our methodology will attract interest from various research communities, and sharing our work at NeurIPS will help achieve this goal. If our response addresses your concerns, we kindly request that you consider raising your score.

---

> > ### Comment · Reviewer_rGnk · 2024-08-13
> > **response**
> >
> > Thank you for the clarifications here and in your above rebuttal. With regards to point 3., I think it would be beneficial to have this demonstration. I will keep the score as it is

---

### Author Rebuttal · Authors · 2024-08-07

We thank the reviewers for their insightful comments and questions. Your feedback provided valuable suggestions and directions for improvement. Below is a brief summary highlighting the main enhancements and changes we will be making.

**Limitation and Generalization**

We will add the following concluding remarks section to clarify the limitations of our theoretical results for the proposed generator gradient (GG) estimator. Additionally, we will address the possibility of extending our theoretical results to prove the consistency of the GG estimator when the smoothness assumptions (Assumptions 1 and 3) are violated.

>The theoretical results in this paper have the limitation of requiring second-order continuous differentiability and uniform boundedness of the space derivatives of the parameters of the underlying jump diffusion. These strong conditions, which are standard in the literature of stochastic flows (cf. [Protter 1992] and [Kunita 2019] cited in the paper) to guarantee the global existence and uniqueness of the derivative processes in (3.5), are necessary to achieve the generality of the results presented in this paper.
>
> However, our generator gradient estimator often works even when coefficients are not continuously differentiable. This is true if the generator and rewards gradients are defined almost everywhere, and the derivative processes in (3.5), with almost everywhere derivatives of the SDE parameters, exist for every $t \in [0, T]$ and satisfy some integrability conditions. Examples include neural networks parameterized stochastic control with ReLU activation functions, heavy-traffic limits of controlled multi-server queues, and the Cox–Ingersoll–Ross (CIR) model. For these models, the existence and integrability of the derivative processes can be checked on a case-by-case basis, allowing the consistency and unbiasedness of the generator gradient estimator to be established. We confirm this by numerically investigating the CIR process and an SDE with ReLu drift in Appendix G.}}

As promised in the concluding remarks, we conducted additional numerical experiments using the CIR process:
$$X_\theta^x(t,s) = x + \int_t^s(\theta - X_\theta^x(t,r))dr + \int_t^s\sqrt{X_\theta^x(t,r)}dB(r)$$
and the ReLU drift SDE:
$$X_\theta^x(t,s) = x + \int_t^s(\mathrm{ReLU}(\theta X_\theta^x(t,r)) + 1)dr + \int_t^s dB(r),$$
where $x, X_\theta^x(t,s), \theta \in \mathbb{R}$. We used the proposed GG estimator and the finite difference derivative estimator (a consistent gradient estimator for both cases, though it performs poorly in high-dimensional settings) to evaluate the gradient $\partial_\theta v_\theta(0,x)$ of
$$v_\theta(0,x) = E\left[\int_0^T X_\theta^x(t)dt\right]$$
evaluated at different $\theta > 0$.

Numerical results in Tables 3 (CIR) and 4 (ReLU) in the attached supplement PDF indicate that in both cases, the GG estimator is consistent even if the global smoothness of the SDE parameters is violated.

However, when $\theta < 1/2$, the CIR process is known to touch 0, making the derivative process in (3.5) only defined up to the first hitting time of 0. Therefore, the previously discussed generalized conditions don't hold. Although the numerical results in Table 3 still show consistency, we observe a surge in variance, suggesting that the GG estimator doesn't work reliably for the CIR case when $\theta < 1/2$.

**Error Analysis of Numerical SDE Schemes**

We believe it would be of significant theoretical value to demonstrate that the error of our estimator scales well with dimensions. However, it is challenging to investigate the convergence property of the algorithm in terms of the dimensions of $\theta$ and $x$. Below, we include a brief error analysis and explain why we decided against including it in the paper.

In our numerical experimentation, we use Euler schemes to approximate the SDE, which introduces a weak error (bias) of $O(\delta)$, where $\delta$ is the time-step discretization (cf. [1]).

Typical analysis of the Euler scheme (also see [1]) results in a strong error of $O(\sqrt{\delta})$. Consequently, the second moment of our estimator using the Euler scheme can be bounded by
$$E|D_\delta(x)|^2 \leq E|D_\delta(x) - D(x)|^2 + E|D(x)|^2 \leq C_1\delta + C_2(|x|+1)^{2m+4},$$
where $D_\delta(x)$ is the GG estimator from SDEs simulated using the Euler scheme, and the bound for the second term follows from Theorem 3. Here, $C_1$ would depend on $T$ and the boundedness of the space derivatives (up to second order) of $\nabla_\theta\mu_\theta$, $\nabla_\theta \sigma_\theta$, $\nabla_\theta\chi_\theta$, $\nabla_\theta\rho_\theta$, and $\nabla_\theta g_\theta$. $C_2$ depends on $T$ and the boundedness of space derivatives (up to second order) of $\mu_\theta$, $\sigma_\theta$, $\rho_\theta$, $\chi_\theta$, and $g_\theta$.

These bounds are typically very loose in real-world examples and do not provide new insights into the validity of the GG estimator. Additionally, rigorously presenting this error analysis would require introducing a significant amount of new notation. Therefore, due to clarity and space limitations, we have decided not to include such an analysis.

[1] Kloeden, P.E., Platen, E. (1992). Numerical Solution of Stochastic Differential Equations. Applications of Mathematics, vol 23. Springer, Berlin, Heidelberg.

---

> ### Comment · Reviewer_5TVX · 2024-08-12
> **CIR at zero**
>
> Thank you for providing this additional numerical study!
>
> I am wondering how to read your results for small $\theta$. If $\theta> 1/2$ the model is essentially smooth as the paths avoid the irregular point zero and the coefficients are smooth away from zero. To check empirically if the estimator also works in non-smooth scenarios you are interested in small $\theta$ to challenge the non-smoothness. The careful interpretation would be the estimator potentially does not work well in non-smooth settings. Azt least in small dimension it seems to be much worse than FD. It could happen (I am only speculating) that for non-smooth coefficients in high dimension the entire advantage of the estimator gets lost in gigantic variances.
>
> The rebuttal statements the estimator has wide applicability also for non-smooth coefficients might be a bit far fetched. Might be true or not, we do not have good evidence in both directions.

---

> > ### Author Response · Authors · 2024-08-12
> > **Clarification**
> >
> > Even though the CIR process will not hit 0 when $\theta> 1/2$, the process can be arbitrarily close to 0 with positive probability. So, this is still a non-trivial case of non-smooth volatility. On the other hand, the ReLU SDE is also one with a non-differentiable drift. In these two cases (CIR with $\theta>1/2$ and ReLU drift), the derivative processes are well defined and integrable. As explained before, our theory can be extended to establish the validity of the generator gradient estimator in these context. We also empirically validate this consistency.
> >
> > However, when $\theta<1/2$, as we have explained, the derivative processes are not globally defined. So, as we have explained, we cannot guarantee the consistency of our estimator. This is reflected by a surge of variance as $\theta\downarrow 0$.
> >
> > Nevertheless, the use case of our proposed methodology usually involves SDEs parameterized by deep neural networks, which are typically smooth (or of ReLU-type) in $X$. Therefore, in these applications, the smoothness should not be a concern to users of our methodology.
> >
> > Moreover, we would like to further assure the reviewer of the validity by mentioning that we have experimented with the neural network parameterized LQ control example using ReLU activation functions. We observe similar computation and variance performance.

---

### Decision · Program_Chairs · 2024-09-25

**Decision:**

Accept (poster)

**Comment:**

The article studies classical stochastic control in the continuous d-dimensional SDE setting and introduces a creative switching trick that reduces the simulation dimension when computing the derivative. After reading all the reviews/rebuttals/responses and the paper, the results are sound and interesting. Nonetheless, there was a significant discussion regarding the applicability and interest to the ML community of the results as the paper is dense and difficult to read with little motivations to ML. While I understand that the authors want this result to be applicable to a broader audience, some of the responsibility lies on the authors to write a paper that also appeals to the broader audience. In not doing so, the paper runs the risk that no one reads it. Please consider this and the discussion with the reviewers in the revision.